# Decision tree-based detection of blowing snow events in the European Alps

Zhipeng Xie[1], Weiqiang Ma[1,3], Yaoming Ma[1,3], Zeyong Hu[2,3], Genhou Sun[4], Yizhe Han[1,5], Wei Hu[1,5], Rongmingzhu Su[1,5], Yixi Fan[1,5]

[1] Key Laboratory of Tibetan Environment Changes and Land Surface Processes, Institute of Tibetan Plateau Research, Chinese Academy of Sciences, Beijing, 100101, China.
[2] Key Laboratory of Land Surface Process and Climate Change in Cold and Arid Regions, Northwest Institute of Eco-Environment and Resources, Chinese Academy of Sciences, Lanzhou, 730000, China.
[3] CAS Center for Excellence in Tibetan Plateau Earth Sciences, Beijing, 100101, China.
[4] School of Atmospheric Sciences, Sun Yat-sen University, 135 Xingang Xi Road, Guangzhou, 510275, China.
[5] University of Chinese Academy of Sciences, Beijing, 100049, China.

*Correspondence to*: Zhipeng Xie (zp_xie@itpcas.ac.cn)

**Abstract.** Blowing snow processes are crucial in shaping the strongly heterogeneous spatiotemporal distribution of snow, and in regulating subsequent snowpack evolution in mountainous terrain. Although empirical formulae and constant threshold wind speed have been widely used to estimate the occurrence of blowing snow in regions with sparse observations, the scarcity of in-situ observations in mountainous regions contrasts with the demands of models for reliable observations at high spatiotemporal resolution. Therefore, these methods struggle to accurately capture the high local variability of blowing snow. This study investigated the potential capability of the decision tree model (DTM) to detect blowing snow in the European Alps. The DTMs were constructed based on routine meteorological observations (mean wind speed, maximum wind speed, air temperature and relative humidity) and snow measurements (including in-situ snow depth observations and satellite-derived products). Twenty repetitions of random sub-sampling validation test with an optimal size ratio (0.8) between the training and validation subset were applied to train and assess the DTMs. Results show that the maximum wind speed contributes most to the classification accuracy, and the inclusion of more predictor variables improves the overall accuracy. However, the spatiotemporal transferability of the DTM might be limited if the divergent distribution of wind speed exists between stations. Although both the site-specific DTMs and site-independent DTM show great ability in detecting blowing snow occurrence and are superior to commonly used empirical parameterizations, specific assessment indicators varied between stations and surface conditions. Events for which blowing snow and snowfall occurred simultaneously were detected the most reliably. Although models failed to fully reproduce the high frequency of local blowing snow events, they have been demonstrated a promising approach requiring limited meteorological variables and have the potential to scale to multiple stations across different regions.

## 1 Introduction

Wind plays a key role in the snow distribution in mountainous terrain, as it shapes both the spatial heterogeneity of snowfall and the erosion, transport and deposition of surface snow via blowing snow processes. In addition to their impacts on the strong spatiotemporal heterogeneity of the surface snow distribution, blowing snow processes also have important consequences on the subsequent evolution of the snowpack (Déry and Yau, 2002; Leonard and Maksym, 2011) and the surface water and energy budgets (Lenaerts et al., 2012a; Liston, 2004; Pomeroy and Gray, 1995; Sexstone et al., 2018). Meanwhile, wind-induced snow transport can also be a major hazard, causing severe reductions to visibility near the ground and triggering snow avalanches (Lehning and Fierz, 2008), with the potential for loss of life, property damage, and disruption of transportation. Blowing snow events result in large-scale snow-mass divergence or convergence from open, wind-exposed surfaces to wind-sheltered areas such as densely vegetated surfaces and topographic depressions (Essery and Pomeroy, 2004). Micro- and mesoscale variability in snow cover and snowmelt strongly influence the surface radiation balance, surface discharge, ecology, and soil freeze/thaw, and can be largely attributed to the spatial heterogeneity of surface snow redistribution caused by blowing snow (Liston, 2004; Mott et al., 2018). Therefore, wind-driven snow redistribution is widely recognized as driving patterns in snow accumulation and snowpack evolution in alpine basins, and represents an important interaction between the land and the overlying atmosphere.

Several specific instruments facilitate direct observation of blowing snow at local scale. For example, the mechanical traps used by Budd et al. (1966), the optical sensors deployed in the Antarctic and Alps (Snow particle Counters, SPC; Sato et al., 1993; Nishimura and Nemoto, 2005; Vionnet et al., 2013), and the acoustic sensors (i.e., FlowCapt and SPC) used to provide reliable measurements of blowing snow mass flux (Chritin et al., 1999; Trouvilliez et al., 2015). However, direct near-surface blowing snow observations are extremely sparse in time and space. Alternative methods using empirical formulae to parameterize blowing snow occurrence have been proposed (e.g., He and Ohara, 2017; Li and Pomeroy, 1997a; Schmidt, 1980). One of the most important parameters is the threshold wind speed for snow transport, as it determines the occurrence of blowing snow. Blowing snow event takes place when the wind exceeds the threshold wind speed. Previous studies have demonstrated that cohesive resistance increases dramatically when snow becomes wet, as the melt water increases the associated cohesion between the particles (e.g., Li and Pomeroy, 1997a; Schmidt, 1980), and sintering of snow particles have a significant bearing on the cohesive force development as well (He and Ohara, 2017; Schmidt, 1980). Therefore, the presence of liquid water and the associated snow metamorphism and aging processes typically increase the bond strength in the surface snow layer (Bromwich, 1988; Li and Pomeroy, 1997a). As summarized by Schmidt (1980), the threshold wind speed highly depends on the cohesion between snow particles and was greatly influenced by temperature, humidity and deposition time.

The threshold wind speed is important for predicting the initialization of a blowing snow event. Threshold wind speed at the height of 10 m was found to be 9.9 m s$^{-1}$ for wet snow and 7.7 m s$^{-1}$ for dry snow, and a formula expresses the threshold wind speed as a function of air temperature has been proposed based on field observations from the Canadian Prairies (Li and Pomeroy, 1997a). Moreover, other parameterizations have also been established, using the relationship between threshold

wind speed and the microstructural properties of surface snow, such as snow density, the bond diameter between snow particles, and the particle mean radius (Gallée et al., 2001; Gallée et al., 2013; Guyomarc'h and Mérindol, 1998; He and Ohara, 2017; Lehning et al., 2000; Schmidt, 1980, 1981). These parameterizations are widely used in numerical models to describe wind-driven snow transport processes. Rather than being constant, it is widely accepted that the threshold wind speed varies with

temperature, humidity, particle size, and deposition time (He and Ohara, 2017). Though there are proposed relationships between the threshold wind speed and meteorological conditions, parameterizations have only been validated for very limited areas (Gallée et al., 2001; Li and Pomeroy, 1997a; Schmidt, 1981) and there is no standard method for determining the meteorological conditions under which blowing snow events occur (Li and Pomeroy, 1997b).

Recent attempts have been made to retrieve blowing snow occurrences from satellite remote sensing data (Palm et al.,

2011, 2018). Results demonstrate the validity of the remote retrieval algorithms in detecting the blowing snow events over the Antarctic, providing insights into the spatial and temporal variability of blowing snow events independently from modeling approaches. The satellite-based technique provides the opportunity to derive blowing snow occurrences with wide spatial coverage, but it is hampered by the presence of clouds and the coarse vertical resolution (Gossart et al., 2017). Moreover, satellite blowing snow detection is associated with pronounced uncertainty and cannot detect the presence of blowing snow

events at fine temporal resolution (Palm et al., 2011), preventing its widespread application in remote areas outside the Antarctica.

Progress has been made in obtaining large spatial scale blowing snow estimates using various multiple data sources, such as visual observations (Mahesh et al., 2003), ground-based ceilometer observations (Gossart et al., 2017), and snow depth and simultaneous meteorological observations (Guyomarc'h et al., 2019; Guyomarc'h and Mérindol, 1998; Vionnet et al., 2013).

However, direct observations are scarce, both in time and space. Snow depth measurements are more common than visual blowing snow observations or ground-based ceilometer observations, but are not routinely included in conventional meteorological observation systems. Meanwhile, numerical modelling provides a useful tool to estimate blowing snow occurrences, but relies not only on accurate forcing datasets (e.g., temperature and wind speed), but also on knowledge of the surface snow properties, which are difficult to accurately define. This is particularly notable in mountainous regions such as

the Alps, where the surface is strongly heterogeneous and environmental conditions are very variable.

Whether a blowing snow event occurs or not is an important state variable for detailed simulations of blowing snow processes. Standard meteorological instruments (distinct from specific instruments such as SPC and FlowCapt which are less commonly deployed) are often used in blowing snow studies. In this study, we use a machine learning based decision tree model (DTM) to detect the presence of blowing snow by exploiting routine meteorological observations (such as wind speed,

air temperature, precipitation and relative humidity) and snow measurements (in-situ snow depth observations and satellite derived products) from 10 ISAW stations (http://isaw.ch/). This study aims to develop a simple but efficient tool to detect blowing snow occurrences and to advance our understanding of the relationships between blowing snow processes and ambient meteorological conditions.

## 2 Data and Methods

### 2.1 Data

Data were obtained from ISAW, and include measurements of blowing snow fluxes and surface meteorological variables. These include mean and maximum wind speed (WS and WSMAX) at 3.5 m height, wind direction (WD), air temperature (T), relative humidity (RH), snow depth (SD), and precipitation about 30 observation stations. The surface meteorological data are measured at every minute, and hourly averages are stored. Although the available meteorological variables vary between stations (for example, RH is only available at Fmor, Fcmb, Fber, Fhue and Fgie), each ISAW station is equipped with the FlowCapt acoustic sensor (Chritin et al., 1999) to measure blowing snow fluxes. In this study, 10 stations that include all the above-mentioned observations were selected (Table 1), of which the Fsal station was used in the sensitivity test and not used in constructing the DTMs. This maximizes the number of different dimensions (corresponding to the different observed fields) that can be used to construct an efficient DTM for identifying blowing snow events.

**Table 1.** List of stations used in this study.

| Station | Latitude (°N) | Longitude (°E) | Elevation (m) | Data period |
|---------|---------------|----------------|---------------|-------------|
| Fmor | 45.02965 | 5.880547 | 2140 | 2013.09.17-2020.12.10 |
| Fcmb | 45.017311 | 6.17845 | 2460 | 2015.11.23-2020.12.10 |
| Fber | 44.949944 | 6.237082 | 2390 | 2013.09.17-2020.12.10 |
| Fhue | 45.10188 | 6.056158 | 2064 | 2013.09.17-2020.12.10 |
| Fgie | 45.855289 | 6.525692 | 1812 | 2011.06.02-2020.12.10 |
| Fmon | 45.30283 | 6.56593 | 2280 | 2013.11.02-2020.12.10 |
| Fche | 45.513248 | 6.95439 | 2869 | 2011.05.16-2020.12.10 |
| Fbon | 45.362431 | 7.05232 | 2480 | 2011.05.16-2020.12.10 |
| Fcel | 45.49057 | 6.40921 | 1924 | 2011.05.16-2020.12.10 |
| Fsal | 44.856754 | 5.952989 | 1975 | 2019.06.19-2020.12.10 |

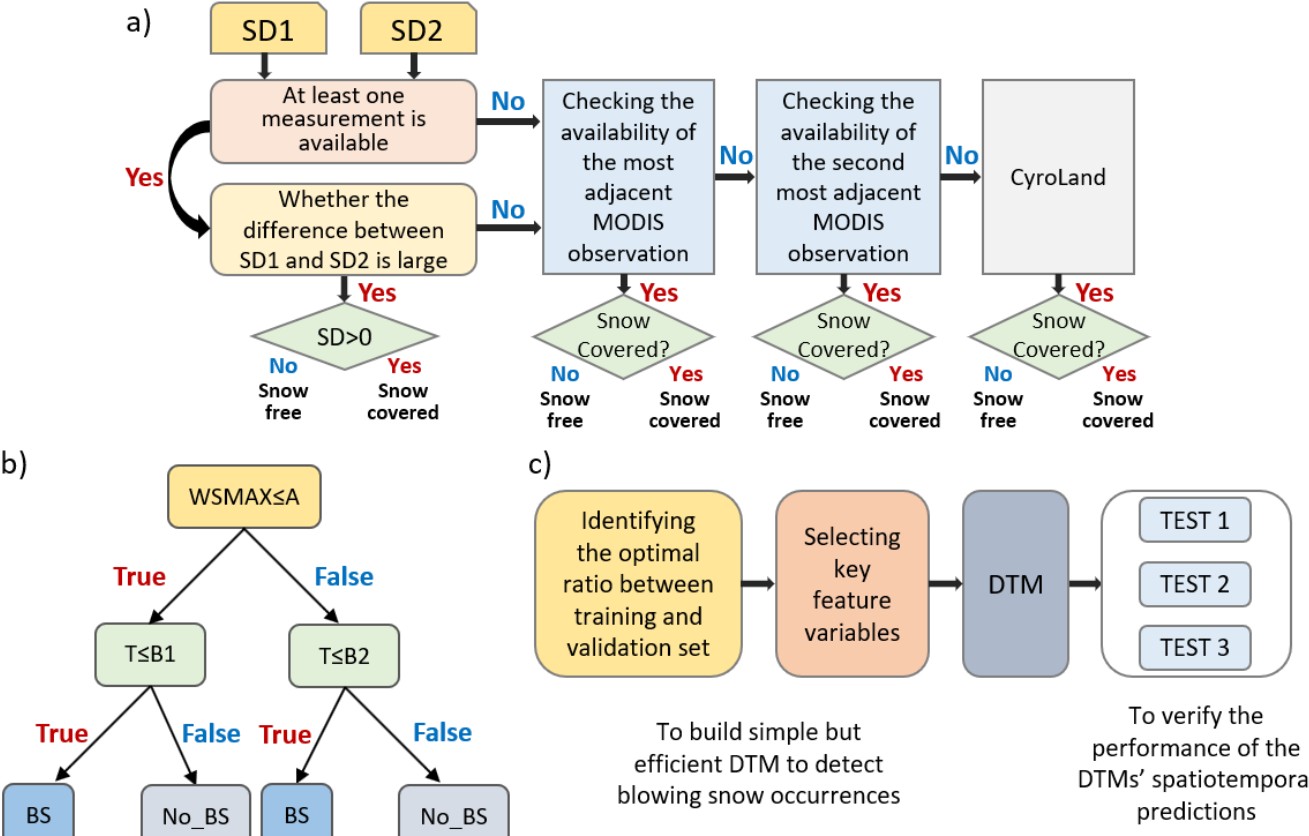

**Figure 1**. Schematic flowchart of a) the procedures to identify the presence of snow, and b) flowchart of a simple decision tree model to detect blowing snow occurrence (only WSMAX and T were used to construct the DTM, A denotes the threshold maximum wind speed, B1 and B2 denote the threshold air temperature), and c) logical framework of this study. BS and No_BS denotes with and without blowing snow occurrence, respectively.

To minimize uncertainty in the classification stemming from the use of poor-quality data, strict data selection criteria were applied to observations. First, using a threshold value of 50% change within an hour, the main change range check was applied to the relative humidity to detect its abnormal change. In addition, a threshold check was performed for the hourly measured air temperature, wind speed, maximum wind speed. For example, data with T outside the range -50 to 50°C or WSMAX greater than 40 m s$^{-1}$ were considered unreliable and were discarded. Periods when both WS and WSMAX were zero for more than three consecutive hours and non-zero WS remain unchanged for more than five hours, were also removed. Since blowing snow fluxes measured by the FlowCapt sensors are sensitive to soil particles, false signals are frequently detected; therefore, only data from winter and spring (from November to April) were used, minimizing the uncertainty resulting from this issue. Additional suspicious data were discarded when a blowing snow event was recorded by the FlowCapt sensor without concurrent snowfall and in the absence of snow cover, or when the positive air temperature lasted for more than 24 hours. In

this study, periods of blowing snow occurrence were identified when positive blowing snow flux was observed. This is different from the work of Trouvilliez et al. (2015) who used a threshold of 1g m$^{-2}$ s$^{-1}$ to remove non-significant blowing snow occurrences and the work of Vionnet et al. (2013) who only analyzed events of duration longer than 4 hours. The presence of snow on the ground was determined based on the snow depth measurements from two snow depth sensors, the MODIS daily snow cover product (MOD10A1 and MYD10A1, Hall and Riggs, 2021a, b) and the CryoLand fractional snow cover product over Alps (http://cryoland.enveo.at). For detailed procedures please refer to the schematic flowchart in Figure 1a.

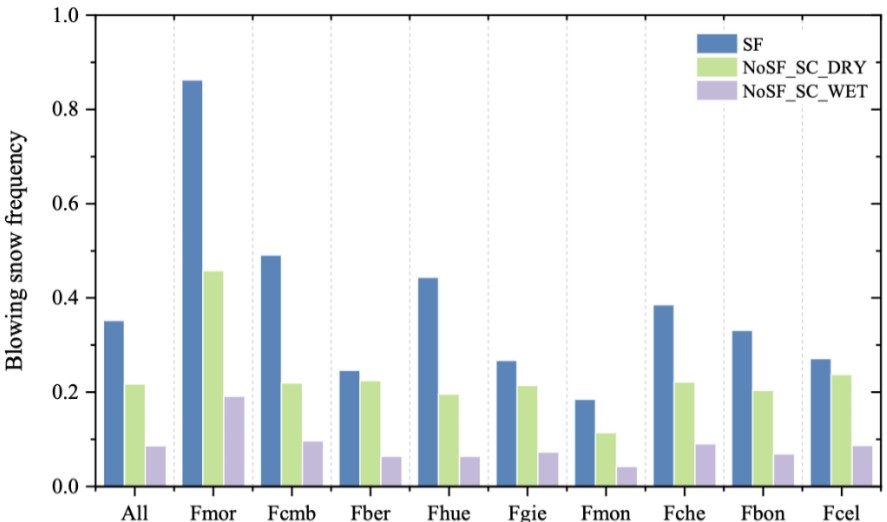

**Figure 2**. The frequencies of occurrence of the 3 types of blowing snow at each station and at all stations combined. SF denotes snowfall condition and NoSF_SC_DRY denotes surface covered by dry snow without concurrent snowfall condition, and NoSF_SC_WET denotes surface covered by wet snow without concurrent snowfall condition. The blowing snow frequency denotes the ratio between occurrences of blowing snow for a given atmospheric condition divided by the total number of occurrences of this atmospheric condition.

As discussed above, internal physical properties of the snowpack, such as snow particle bonding, cohesion, and its kinetic properties, greatly influence the strength of snow resistance, which determines the initiation and persistence of blowing snow events (Li and Pomeroy, 1997b; Pomeroy and Gray, 1990; Schmidt, 1980). Previous studies have shown a sharp contrast in the threshold wind speed for snow transport between fresh snow and aged snow (Huang et al., 2008; Liston et al., 2007; Pomeroy 2005; Xie et al., 2019), owing to the strength of the bond between snow particles depending on destructive metamorphism, melting, snow loading, and increased compaction caused by overburden (Li and Pomeroy, 1997a, b; Oleson et al., 2010). Newly fallen snow particles are characteristically soft and powdery, with relatively low density, making new snow particles much more likely to be lifted by the wind. The occurrence of snowfall within the hourly measurement interval is therefore the primary factor used to distinguish a blowing snow event in the classification samples. Snowfall occurs when a precipitation greater than 0 coincides with an air temperature ≤0°C. As cohesive resistance increases dramatically once snow

becomes wet, there are very considerable differences in the frequencies of blowing snow occurrence in dry snow and wet snow (Li and Pomeroy, 1997b). Thus, the wet/dry snow condition is also used as an attribute in establishing the classification tree model. Wet snow refers to the snow which has either melted or received liquid precipitation since the last snowfall, while dry snow defines as snow that has not received temperatures of 0 °C or above, or liquid precipitation (Li and Pomeroy, 1997a). To accurately capture the different effects of ambient atmospheric conditions on the occurrence of blowing snow, the quality-controlled data were categorized into three types: snowfall (SF), surface covered by wet snow without concurrent snowfall (NoSF_SC_WET), and surface covered by dry snow without concurrent snowfall (NoSF_SC_DRY). The occurrence frequencies of the 3 types of blowing snow at each station and at all stations combined are shown in Figure 2. A higher occurrence of blowing snow events was detected under concurrent falling snow condition than other conditions.

## 2.2 Method

Decision tree analysis uses a machine learning algorithm to build a tree-like classification structure and regression model to identify a set of characteristics that can best differentiate between individual classes based on a categorical feature variable. This method has become increasingly popular in industrial applications and scientific research. It has been proven to be a very useful and efficient technique in processing remote sensing images (Yang et al., 2017), predicting natural hazard events (Park and Lee, 2014; Ragettli et al., 2017), and estimating hydro-meteorological variables such as winter lake ice (Sharma et al., 2019) and snow depth (Gharaei-Manesh et al., 2016).

**Table 2.** Summary of three tests conducted to verify the performance of the spatiotemporal prediction of the DTM.

|  | Test 1 | Test 2 | Test 3 |
|---|---|---|---|
| Training | 80% of the observations of each station; 80% of the observations from all stations. | Observations from five stations with homogeneous distribution of feature variable (Fmor, Fcmb, Fmon, Fbon. Fche) | Observations from five stations with heterogeneous distribution of feature variable (Fmor, Fcmb, Fmon, Fsal, Fcel) |
| Validation | 20% of the observations of each station; 20% of the observations from all stations. | Observations from Fhue, Fber, Fgie, Fcel, Fsal | Observations from Fhue, Fber, Fgie, Fon, Fche |

A classification tree is composed of decision nodes that represent attributes of the samples to be classified, branches that represent the different possible outcomes of a decision node, and leaves that represent the possible classification (Figure 1b shows a simple decision tree model). Building a classification tree requires two steps: a learning step and a classification step. In the learning step, a classification model is developed based on multi-dimensional training data with labeled attributes. In this step, the maximum depth of the tree can be used as a control variable for pre-pruning to optimize the decision tree. In the classification step, independent data are used to verify the accuracy of the constructed model.

In this study, we use the scikit-learn package (Pedregosa et al., 2011), an open-source Python module for machine learning, to build DTMs and to identify the occurrence of blowing snow events based on routine meteorological observations. Information on the construction of the decision tree model is detailed in the next section, for instance, the selection of key characteristic variables used to build the tree and identification of the optimal ratio between the training and validation set. We conducted three sensitivity tests to verify the spatiotemporal prediction of the established DTMs (Table 2). In Test 1, 80% of the out-of-bag (OOB) observations from each site and all stations were used to construct the site-specific decision tree model (SSDTM) and site-independent decision tree model (SIDTM). This test offers a comprehensive assessment of the decision tree model in identifying the occurrence of blowing snow at both temporal and spatial scales. In Test 2 and Test 3, only five stations were selected to train the model. The main difference between these two tests lies in the distribution characteristics of the feature variables among stations, three stations (Fmor, Fcmb and Fmon) were both involved in these two tests. Tests 2 and 3 serve as a complementary test for further accuracy assessment of the spatiotemporal prediction. The logical framework of this study is presented in Figure 1c.

**Table 3.** Contingency table for blowing snow events and the corresponding indices used for the computation of evaluation metrics.

| | | Observed blowing snow event | |
|---|---|---|---|
| | | Yes | No |
| Estimated blowing snow event | Yes | a | b |
| | No | c | d |

To reduce the classification uncertainty attributable to training data selection, twenty repetitions of a random sub-sampling validation method were applied (with the optimal ratio between the training and validation set) in the construction of each decision tree model. In each cross-validation, the vast majority of available observations were used for training, and the remaining set was used to validate the model. At the end of the cross-validation, 20 testing probabilities were created, and averaged before the final analysis. The accuracy of the model was first calculated by comparing the actual and predicted classifications. The correspondence of predicted and observed blowing snow events was then quantitatively assessed using the overall accuracy (OA) index, false alarm rate (FAR), probability of detection (POD), Heidke skill score (HSS) and missing rate (MR). These evaluation metrics are defined from the contingency table of dichotomous events in Table 3, and can be written in the form:

$$OA = (a + d)/(a + b + c + d), \tag{1}$$

$$FAR = b/(a + b), \tag{2}$$

$$POD = a/(a + c), \tag{3}$$

$$HSS = \frac{2(ad - bc)}{(a+c)(c+d)+(a+b)(b+d)} \tag{4}$$

$$MR = c/(a + c), \tag{5}$$

The overall agreement between estimated and actual blowing snow events is captured in OA, which ranges from 0 to 1, with 1 representing a perfect classification. The FAR measures the fraction of forecasted events that did not actually occur and the MR denotes the proportion of blowing snow events that actually occurred but not captured by the DTM model (both range from 0 to 1, with optimal performance of 0), and the POD is the fraction of observed blowing snow events that were correctly identified by the models (range from 0 to 1, with 1 representing the perfect score). When one category is dominant, previous studies have reported that the OA is not sufficient, as it can be hedged by forecasting common events more frequently (Roebber et al., 2003; Notarnicola et al., 2013). The HSS accounts for this bias by characterizing the skill of the compared dataset with regards to the no-skill random forecasts. The HSS ranges from -1 to 1, with 1 representing a perfect classification skill, 0 representing a random classification and negative values corresponding to a decision tree-based classification that is less accurate than a random classification.

## 3 Results

### 3.1 Sensitivity to the proportion of training samples

Training a model is the first step in making good predictions. Splitting the available dataset into a training portion and a validation portion is therefore necessary to build a solid basis with which to train and test a model. Theoretically, the DTM should be trained on a larger portion of the data, to more accurately capture the underlying spread and pattern of data. However, real datasets are often imbalanced, with random noise; therefore, a certain portion of the validation dataset must be retained to verify the model's ability in deriving the underlying pattern of observations and to ensure the reliability of the assessment.

**Table 4.** Performance statistics of decision tree model constructed under different situations with varied ratio for training and testing data sets (ranging from 0.5 to 0.9)

| Condition* | Sample size | Range $(10^{-3})$ | Standard deviation $(10^{-4})$ | Max/Min overall accuracy | Description |
|---|---|---|---|---|---|
| SF | 9251 | 7.19 | 15.26 | 0.855/0.848 | Snowfall |
| NoSF_SC_DRY | 74915 | 2.26 | 4.66 | 0.871/0.869 | No snowfall but with dry snow cover |
| NoSF_SC_WET | 258872 | 1.06 | 2.23 | 0.935/0.934 | No snowfall but with wet snow cover |

*: The situations are list based on sample size.

Sensitivity tests were conducted and evaluated to confirm the appropriate ratio between the training subset and validation
subset. Starting with observations from all stations (All), a varying proportion of these stations was retained as the training
dataset (range from 0.5 to 0.9, at 0.01 increments), from which 2400 DTMs were established (here, 2400=40×3×20: this
represents the 40 training sets from 0.5 to 0.9; three groups SF, NoSF_SC_DRY and NoSF_SC_WET; and twenty repetitions
of the random sampling). As listed in Table 4, the overall classification accuracy of these models ranged from 0.848 to 0.935,
indicating that models predicted the occurrence of blowing snow events very accurately. The variation range and standard
deviation of overall accuracy changed slightly with decreasing sample size: the accuracy range ranged from $1.06×10^{-3}$ to
$7.19×10^{-3}$ and the standard deviation increased from $2.23×10^{-4}$ to $15.26×10^{-4}$ as sample size decreased. According to the
sensitivity analysis, training sample size had a small influence on the classification accuracy of the DTM. However, because
the reliability of the accuracy assessment decreased with decreasing validation sample size, the training set proportion of 0.8
was recommended in this study.

**3.2 Sensitivity to the feature variables**

The current decision tree uses a greedy algorithm, meaning that an optimal node construction and attribute combinations
were selected to build the classification tree model. To construct an efficient model capable of being applied across broad
spatial and temporal scales, a major challenge is to select the fewest number of feature variables to construct a model with the
highest classification accuracy. Although many factors (including land surface characteristics and ambient meteorological
conditions) can influence the occurrence of a blowing snow event, it is unrealistic to consider all factors in the estimation,
because of the requirement for spatiotemporal transferability. Therefore, the possible predictor variables used in this study
comprise WS, WSMAX, T and RH.

To assess the relative importance of each single feature variable and to determine a suitable rule for training samples,
nine combinations of feature variables for each of the three conditions (SF, NoSF_SC_WET and NoSF_SC_DRY) were
245 selected from each station and all stations (All) to train the decision tree model. That is to say, in theory, there will be a total
of 5400 DTMs trained (9×10×20×3). However, the RH observations are only available at Fmor, Fcmb and Fber stations (RH
observations from Fhue and Fgie stations were discarded due to their frequent and dramatic fluctuations over short periods);
therefore, 4680 DTMs were eventually obtained. Figure 3 displays the accuracy of these DTMs in estimating the blowing snow
occurrence in the validation samples. The total score, which is a synthetic demonstration of the 20 random sampling tests of
250 the DTMs, is also shown in Figure 3. The total scores are calculated by ranking the accuracies of DTMs trained at each station
with different combinations of predictor variables (the model with the highest accuracy scores the highest: 9 for All, Fmor,
Fcmb and Fber; and 7 for other stations without RH observations). Furthermore, as the mean accuracy was also included, the
maximum values of scores are 189 and 140 for stations with and without RH observations, respectively. As shown in Figure
3, even though the same attribute combination was used, model accuracy varied widely not only between stations but also
between different snow conditions. Generally, models presented higher overall accuracies under dry snow conditions than the

other two conditions. Although models derived for snowfall conditions produced accuracies comparable with models for the ground surface covered by wet snow, the latter were more effective in accurately detecting blowing snow occurrence than the former.

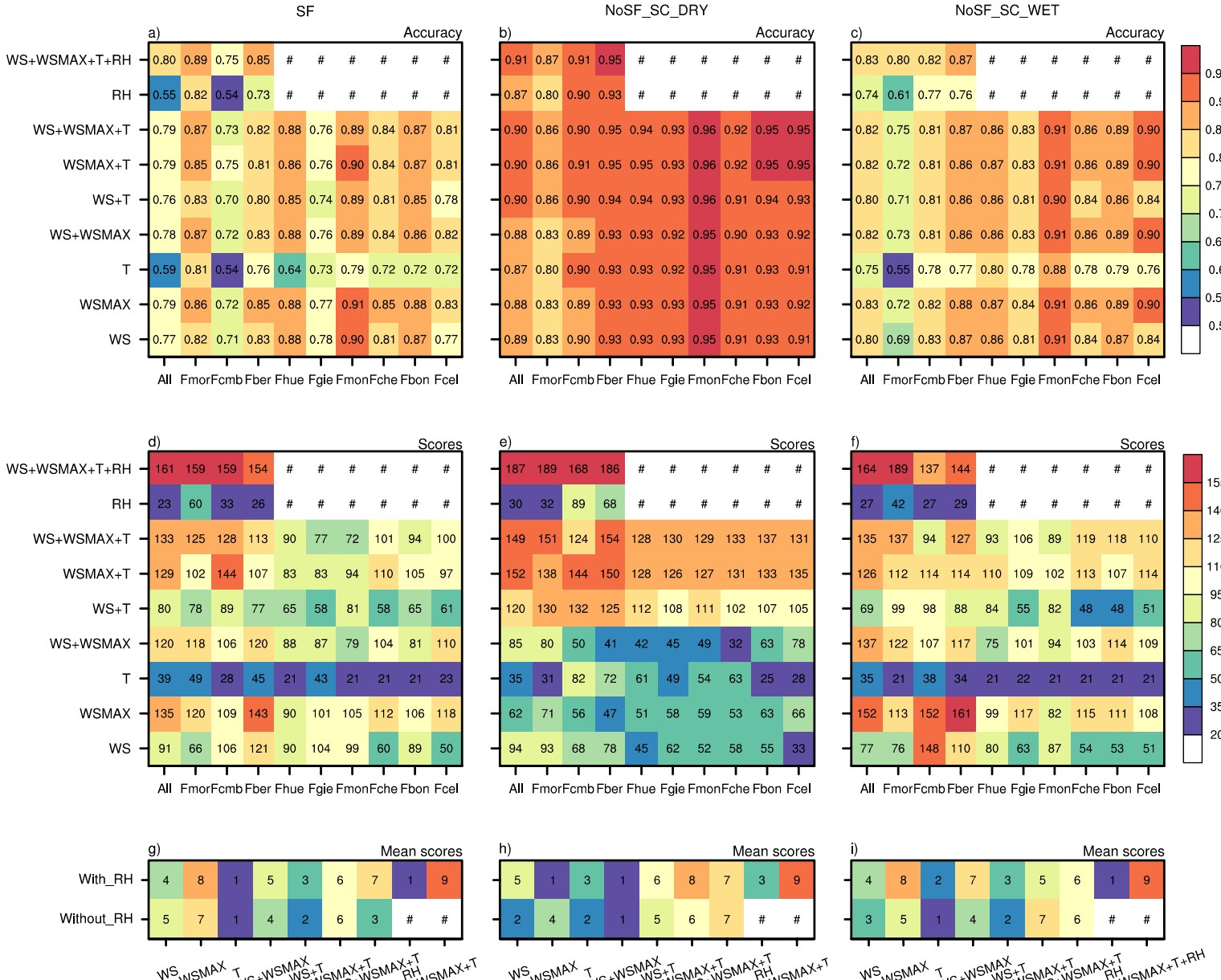

**Figure 3**. The mean overall accuracy (a-c) and scores (d-f) of the site-independent DTM (All) and the site-specific DTMs trained with different feature variable combinations in snowfall conditions (SF), dry snow cover conditions (NoSF_SC_DRY) and wet snow cover conditions (NoSF_SC_WET), and the mean scores of models constructed based on

feature variable combinations with and without the RH included (g-i). The pound sign (#) indicates that the SSDTM was not constructed using RH. WS is mean wind speed; WSMAX is the maximum wind speed; T is air temperature; RH is relative humidity.

### 3.2.1 Air temperature and relative humidity

Of all the attribute combinations evaluated, models trained merely with either T or RH presented the lowest accuracy (Figure 3a-c), indicating that the use of T or RH alone cannot fully capture the variance in the validation samples. However, significant improvements were achieved when either WS or WSMAX were accompanied by T or RH, even though the single factors performed poorly when used alone. Taking Fcmb station in snowfall conditions as an example, model accuracy increased from 0.54 when the model trained merely with T to 0.8 or 0.82 when WS or WSMAX was added, respectively. These results suggest that neither T nor RH is the guarantee of model accuracy, although the model with more predictor variables used generally achieved relatively high accuracy.

### 3.2.2 Mean and maximum wind speed

Models trained with a combination that included WAMAX outperformed the other models, revealing that WSMAX rather than WS contributed the most to the model accuracy, highlighting the importance of WSMAX in constructing a reliable DTM. The result is reasonable, as the fastest wind speed acts as the primary driving force that allows wind shear stress to overcome snow cohesion, bonding and frictional resistance (He and Ohara, 2017). Wind transport of snow can be initiated once the fastest wind speed exceeds the threshold wind speed, and the blowing snow process can then be sustained by relatively low wind speeds. In other words, the fastest wind speed and the mean wind speed control the occurrence and persistence of blowing snow events, respectively. Generally, model accuracy improved as more predictor variables were used. However, strongly correlated feature variables might slightly affect the model accuracy; this was evident when WS was added to WSMAX in snowfall conditions. Overall, this comparison indicates the superiority of DTM as a means of blowing snow identification, which is achieved by making full use of all available feature variables.

The model accuracy analyzed above presents the overall performance of models in identifying the occurrence of blowing snow. Synthetic scores displayed in Figure 3d-f show that the attribute combinations including WSMAX generally achieved higher ranking. Model scores in either wet snow cover or dry snow cover further indicated the need for assimilating more attribute information to improve the classification accuracy of models, particularly in wet snow conditions. These results are closely consistent with the mean overall accuracy results discussed above, implying that they are representative of the mean overall accuracy. Moreover, synthetic scores could reveal further information the averaged accuracy cannot clearly illustrate. For instance, except for the models trained with only T or RH, there were no notable differences in mean accuracy among the various combinations. However, a substantial divergence of synthetic scores was seen among different combinations. In snowfall conditions, WSMAX scored the highest at most of the stations, while the combination of WS, WSMAX and T ranked

highest in the other two conditions. When the land surface was covered by dry snow, the model scores were greatly increased with the inclusion of T, demonstrating the key role of T in influencing the blowing snow occurrences. The contribution of T was more important to the model accuracy than WS or WAMAX, as the blowing snow occurrence regime of dry snow is more sensitive to temperature variation than to shear stress (Li and Pomeroy, 1997b). The scores shown in Figure 3g-i are the integrated scores across stations. One noticeable distinction between models trained with and without RH inclusion was in the optimal attribute combination. When RH was included in the feature variables, the models trained with the combination of WS, WSMAX, T and RH yielded the highest score across all conditions, while the optimal combination varied with conditions when RH was unavailable. This comparison suggested that redundant information lied in the combinations of WS, WSMAX and T might slightly weaken the efficiency of the DTM, and further demonstrated the importance of WSMAX in constructing an accurate and efficient DTM.

Finally, models were trained with only WSMAX in snowfall conditions, while the combination of WS, WSMAX, T and RH (if available) was used for further analysis in snow-covered conditions.

**3.3 Validation of the SSDTMs and SIDTM**

In this section, validations were conducted to assess the predictive performance of DTMs. Similar to the sensitivity tests conducted above, 20 DTMs were trained at each station, based on randomly-selected datasets comprising 80% of the sample observations. The optimal combinations of feature variables determined in Section 3.2 were used. Each DTM was evaluated by the remaining 20% observations at the corresponding training station, and by all available observations from other stations. The divergence between the SSDTMs and SIDTM was compared for assessing the potential of the DTM to be scaled to multiple stations across regions. There were 660 (11×20×3) and 240 (4×20×3) DTMs in total, trained with and without RH, respectively. The mean OA, FAR, POD and HSS of the 20 random sampling tests are compared and analyzed in this section.

According to the results in Figure 4a-c, either the SSDTMs or the SIDIM exhibited high overall accuracy throughout the range of conditions (from 0.66 to 0.96 and 0.72 to 0.96 for models trained with and without RH considered, respectively). However, there were differences among stations and conditions. The DTMs trained and evaluated with observations in wet snow conditions showed the best predictive performance among three conditions except for the evaluations conducted at Fmor station. Overall classification accuracy at stations where SSDIMs were trained (those lying in the diagonal rising up to the right) was not always higher than the accuracy evaluated at other stations, demonstrating the high ability of DTMs in accurately capturing the blowing snow occurrences outside the training range (both temporal and spatial). When comparing the overall accuracy between models trained with and without RH at those stations with RH observations available (Figure 4d-i), the inclusion of RH is seen to increase the performance of DTMs to a certain extent.

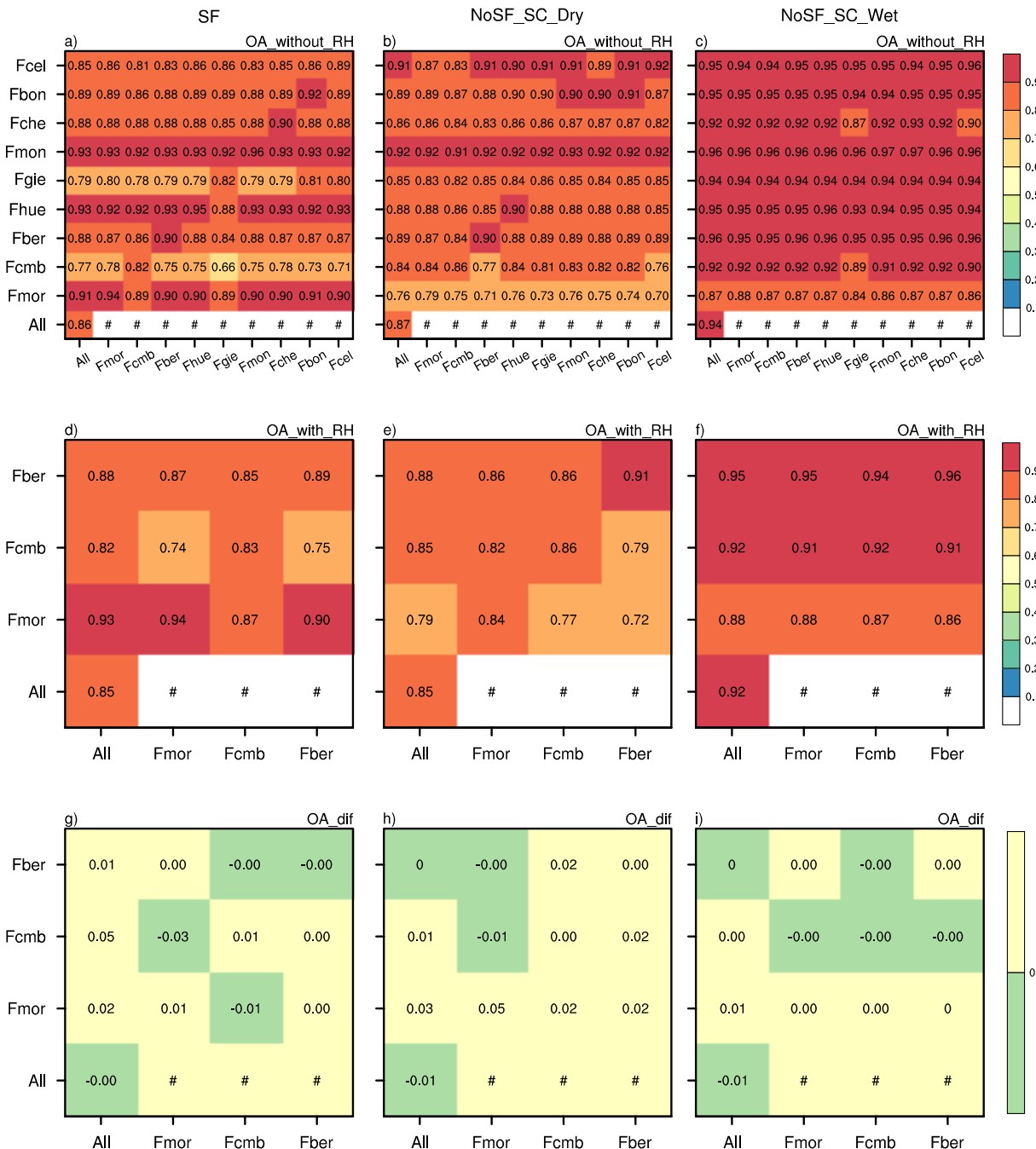

**Figure 4.** The overall accuracy (OA) of SIDTM and SSDTMs trained without (a-c) and with (d-f) RH, and their difference (g-i) in SF, No_SF_DRY and No_SF_WET conditions, respectively. The x axis represents the SIDTM and SSDTMs constructed based on 80% of corresponding observations, and the y axis represents the validation stations.

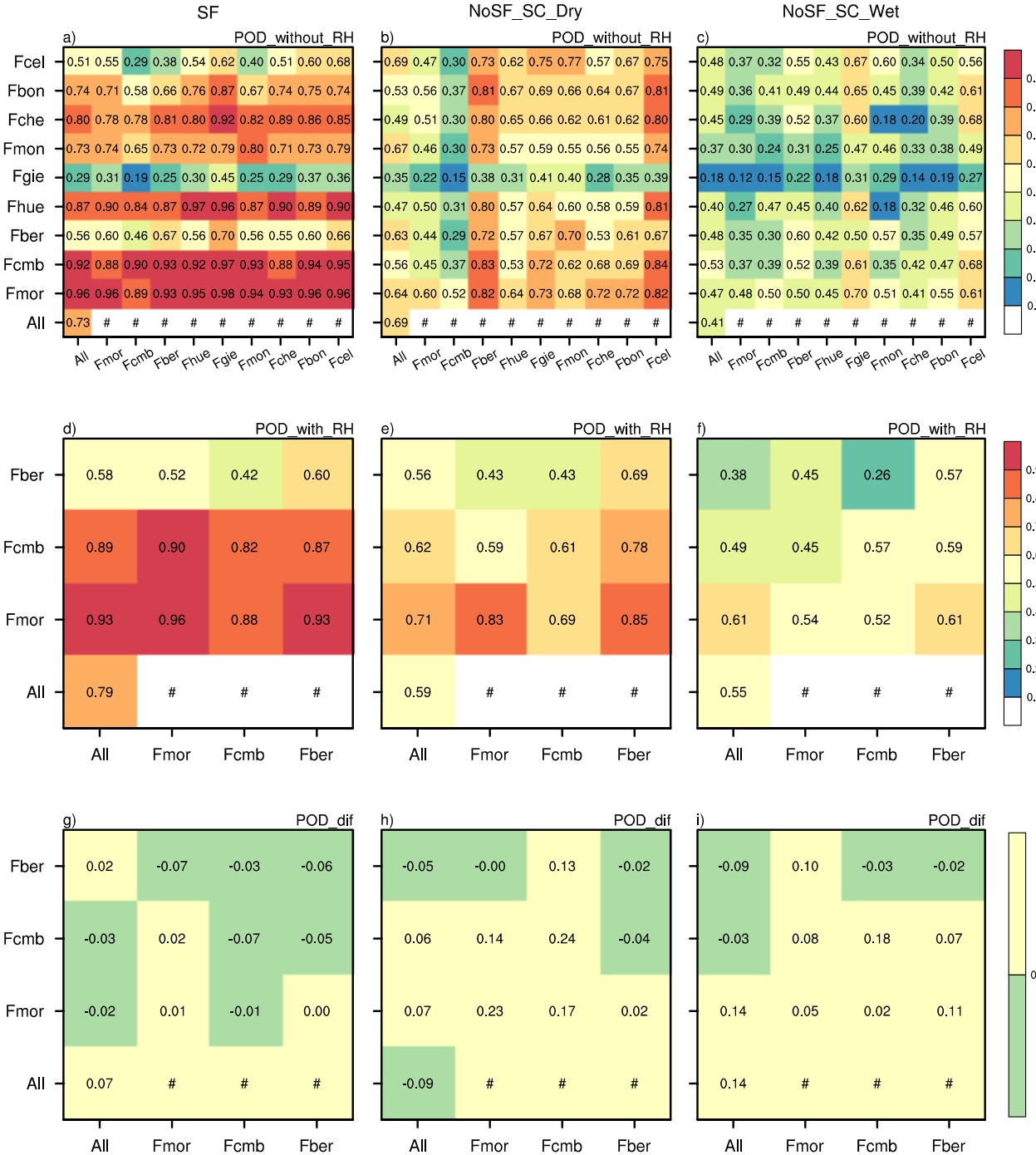

**Figure 5.** The same as Figure 4, but for the probability of detection (POD).

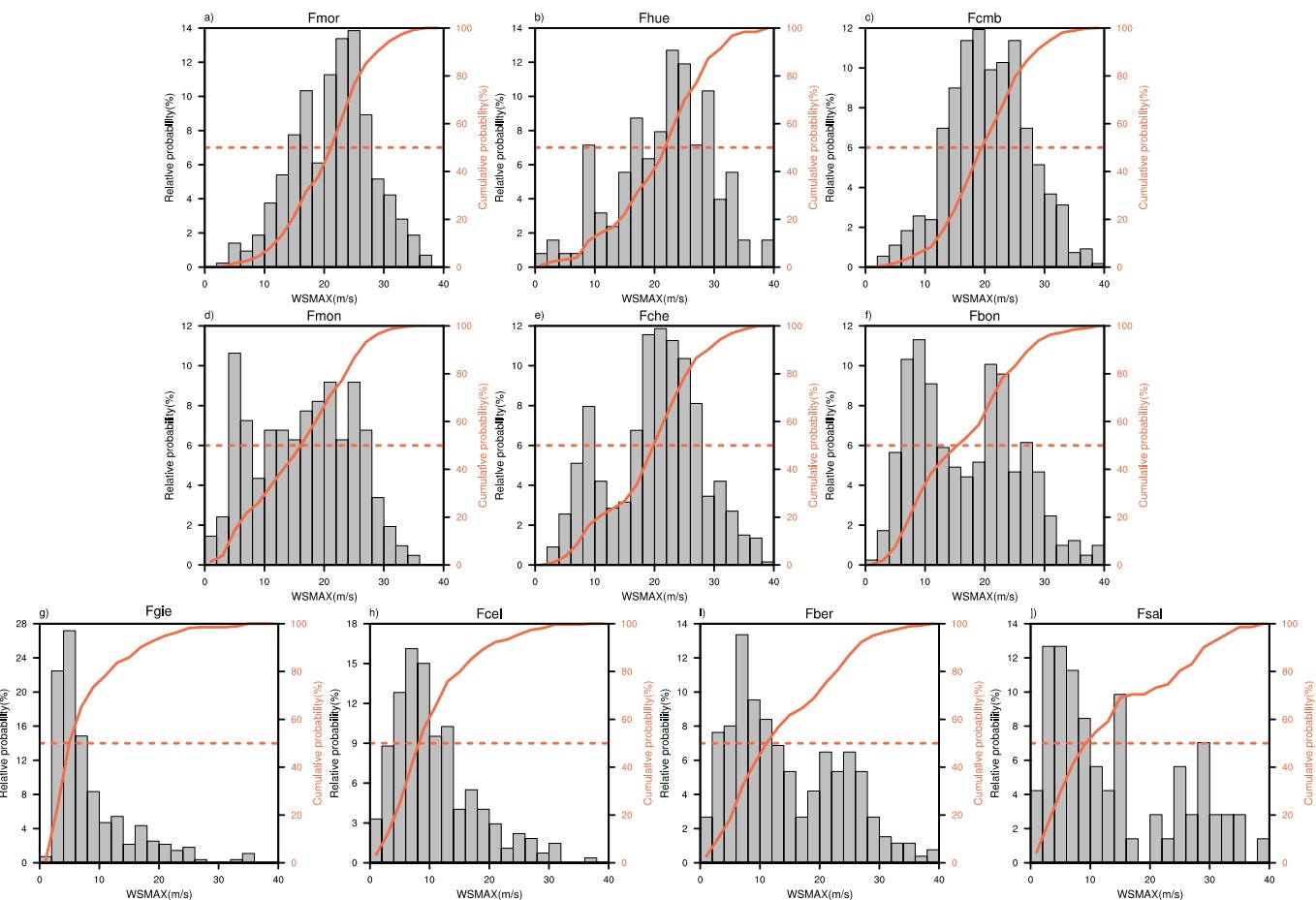

**Figure 6.** The relative probability and cumulative probability of blowing snow occurrences, plotted with the maximum wind speed. The histogram is the relative probability and the orange solid curve is the cumulative probability, respectively. The orange dash line is the 50% cumulative probability.

Although OA represents the overall classification skill of the models, POD is an important metric characterizing the models' ability to detect blowing snow events. As shown in Figure 5, the DTMs constructed under snowfall conditions were generally more accurate in detecting blowing snow occurrences than the models established under the other two conditions, and DTMs exhibited the lowest detecive capacity under wet snow conditions. However, there were clear differences for between both the SSDTMs and SIDTM in detecting the blowing snow events that occurred at different stations. At Fmor, Fcmb and Fhue stations, all DTMs showed consistently high skill in accurately identifying the true blowing snow events (the POD values were above 0.82), even higher than identifying the blowing snow events that occurred at the stations used for training the DTMs. Taking the SSDTM trained at Fgie station as an example, 98%, 97% and 96% of blowing snow events were accurately detected at Fmor, Fcmb and Fhue stations, dropping to 45% at Fgie station (Figure 5a). In snowfall conditions, the POD values dropped sharply when DTMs were evaluated at Fgie and Fcel stations (only up to 45% and 68% of blowing snow events occurring at Fgie and Fcel stations were accurately identified, respectively). Although these differences narrowed dramatically in snow cover conditions (Figure 5b and 5c), the capability of both the SSDTMs and SIDTM remain relatively low in detecting the blowing snow events that occurred at Fgie station; the performance deteriorated further when the surface was covered by wet snow (only up to 29% of blowing snow events were detected). The low POD values corresponding to the high miss rate, indicate that blowing snow events occurring at Fgie station were seriously underestimated by both the SSDIMs and the SIDIM. The differences in POD shown in Figure 5g-i illustrate that the detective ability can be improved when RH serves as a feature variable to train the DTM, as was particularly noticeable in snow cover conditions (with a maximum increase of 24% and 18% for dry snow cover and wet snow cover, respectively). For snowfall conditions, however, the improvement was more limited and was only achieved in 5 of 13 tests (Figure 5g).

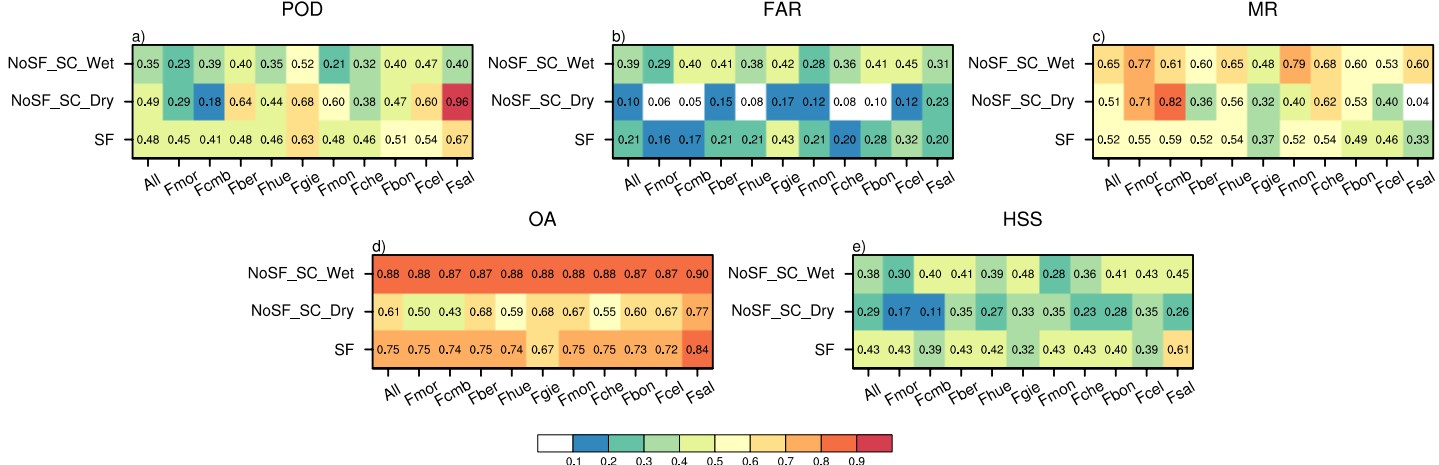

**Figure 7.** The POD (a), FAR (b), MR (c), OA (d) and HSS (e) of SIDTM and SSDTMs assessed at Fsal station, where the maximum wind speed was relatively low (similar to Fgie, Fcel and Fber station).

The relatively low PODs at Fgie, Fcel and Fber station under snowfall conditions reflect the significantly lower maximum wind speeds at these stations when compared with other stations. The relative probability and cumulative probability of maximum wind speed at each station, with concurrent blowing snow (Figure 6), indicate that about 50% of the blowing snow events occurred at Fgie, Fcel and Fber stations occurred when maximum wind speed was below 10 m s$^{-1}$ (Figure 6 g-i), demonstrating that most of the blowing snow events at these three stations were initiated by relatively low maximum wind speed; this was much lower than that at other stations (Figure 6 a-f). DTMs trained with higher maximum wind speeds generally choose a larger threshold WSMAX for the occurrence of blowing snow, thus resulting in an underestimation of blowing snow events when the models were applied to stations with low maximum wind speed. To verify this speculation, DTMs were assessed at Fsal station, since relatively low maximum wind speeds were also reported at this station. As expected, all the DTMs significantly underestimated the frequency of blowing snow events at this station (Figure 7). However, in snow cover scenarios, and except for Fgie and Fsal stations, the distribution of maximum wind speeds was broadly consistent among the stations (Figure not shown), and thus the differences between the models evaluated at these stations were small (Figure 7).

The FAR values evaluated at each station were compared in Figure 8. In general, FAR was slightly lower when falling snow was detected than that in other conditions. For example, except for the model trained at Fgie station, the FAR values for both the SSDTMs and SIDTM evaluated at Fmor, Fhue and Fbon stations were below or equal to 10%, accompanied by high OA and POD values, indicating a good retrieval performance in estimating blowing snow occurrence. However, the blowing snow events under wet snow conditions were more likely to be falsely identified. One obvious distinction in POD values between the SIDTM and SSDTMs trained with RH, was that the SIDTM showed a robust improvement in reducing the probability of false detection whatever the circumstances, while its effectiveness for SSDTMs varied between stations and conditions.

Although blowing snow events frequently occur in the study region, blowing snow is still a rather rare weather phenomenon. Analysis of OA, POD and FAR demonstrated the great ability of the DTMs in accurately identifying the occurrence of blowing snow; however, when considering the impact of the imbalanced dataset, the HSS index, which is particularly suitable for the evaluation of forecasts skill for rare events (Doswell et al., 1990), should be analyzed. Despite the considerable fluctuations in HSS across stations and conditions as shown in Figure 9, the generally high HSS indicated that the DTMs showed promising agreement with in-situ blowing snow measurements (except for Fgie station, where HSS values less than 0.4 were frequently observed). Generally, the consistency between the actual blowing snow events and estimates using the DTM was improved when RH was included in the DTM training. In conclusion, even though the datasets inevitably suffer from imbalanced observations, both the constructed SIDTM and SSDTMs are promising in detecting blowing snow occurrence with considerable accuracy.

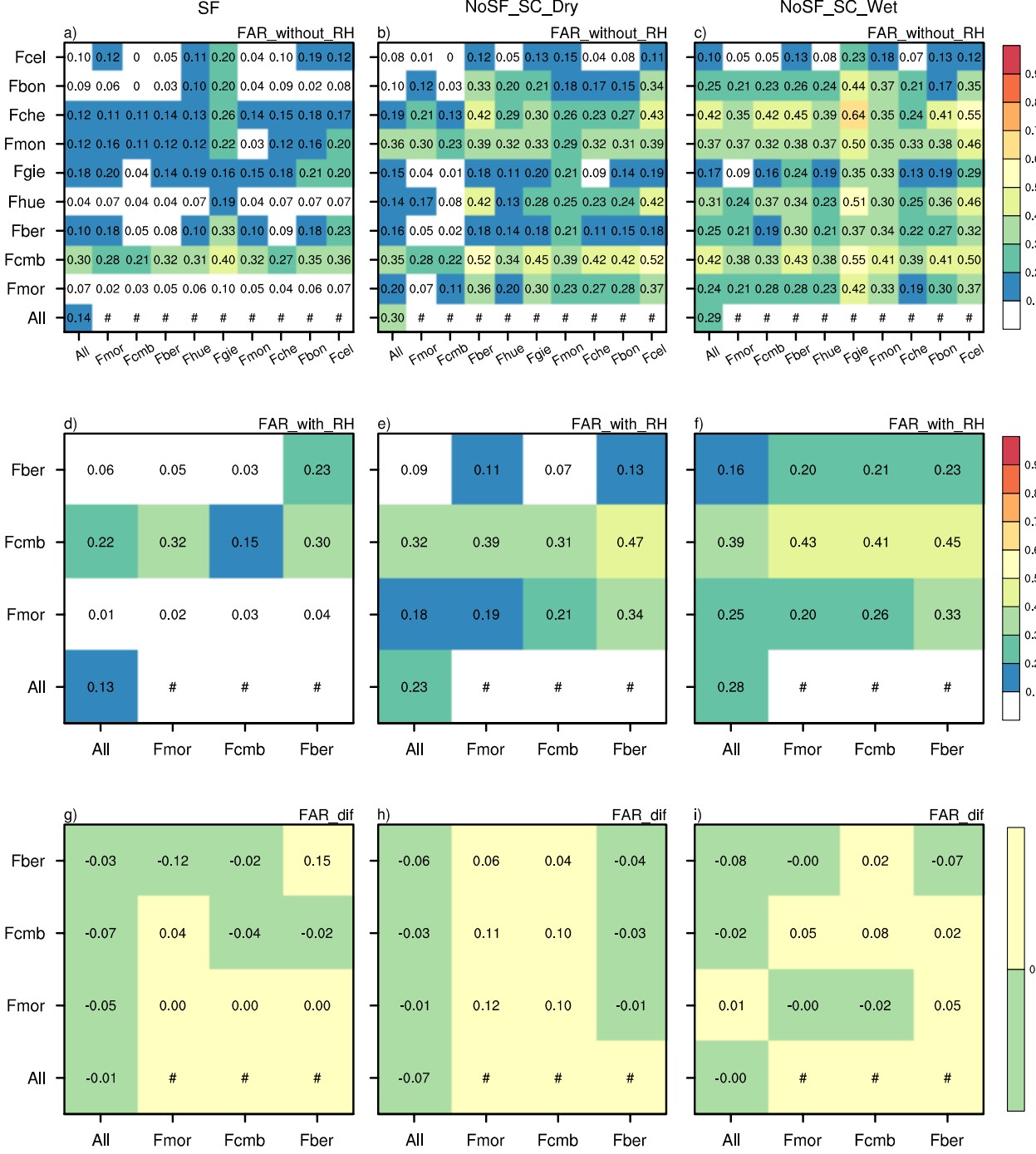

**Figure 8.** The same as Figure 4, but for the false alarm rate (FAR).

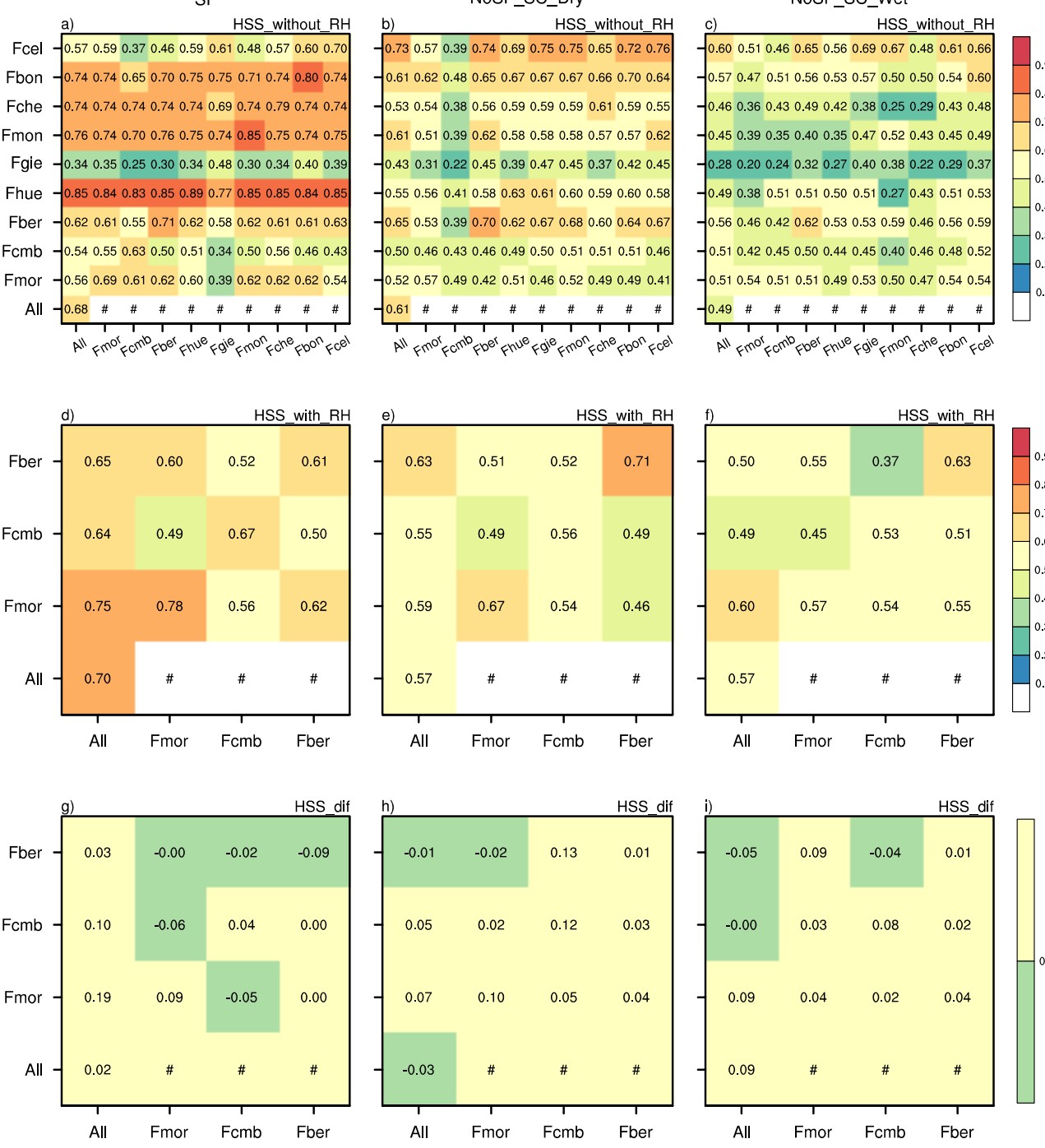

**Figure 9.** The same as Figure 4, but for the Heidke skill score (HSS).

### 3.4 Spatiotemporal transferability assessments

Spatial and temporal transferability refers to how applicable a classifier model is across broad spatial and temporal scales. Due to the temporal independence of the randomly chosen 80% of samples, the accuracy of the temporal prediction of models can be assessed based on the remaining set. Furthermore, the spatially independent datasets from other validation stations can be used to assess the accuracy of spatial prediction. As mentioned in the previous analysis, the constructed DTMs performed commendably in temporal extrapolation, and are very much applicable to the estimation of blowing snow occurrence at other stations outside the training range. In this section, sensitivity tests were conducted to further evaluate and explore the spatiotemporal transferability of the DTMs.

As shown in the assessment of SSDTM across different stations, all SSDTMs showed consistently accurate performance in estimating the occurrence of blowing snow events at the station the SSDTM trained, demonstrating the high capacity of models in the temporal prediction of blowing snow events. Meanwhile, the model accuracy obtained at each validation station was comparable to, or even better than, that assessed at the respective station where each model was trained at. For example, the model trained at Fcmb station yielded an overall classification accuracy of 0.82 when evaluated at Fcmb station (Figure 4a), but the model achieved a markedly higher accuracy when applied at Fmor (0.89), Fhue (0.92) and Fmon (0.92) stations. Notably, this pattern was more pronounced in HSS (Figure 9).

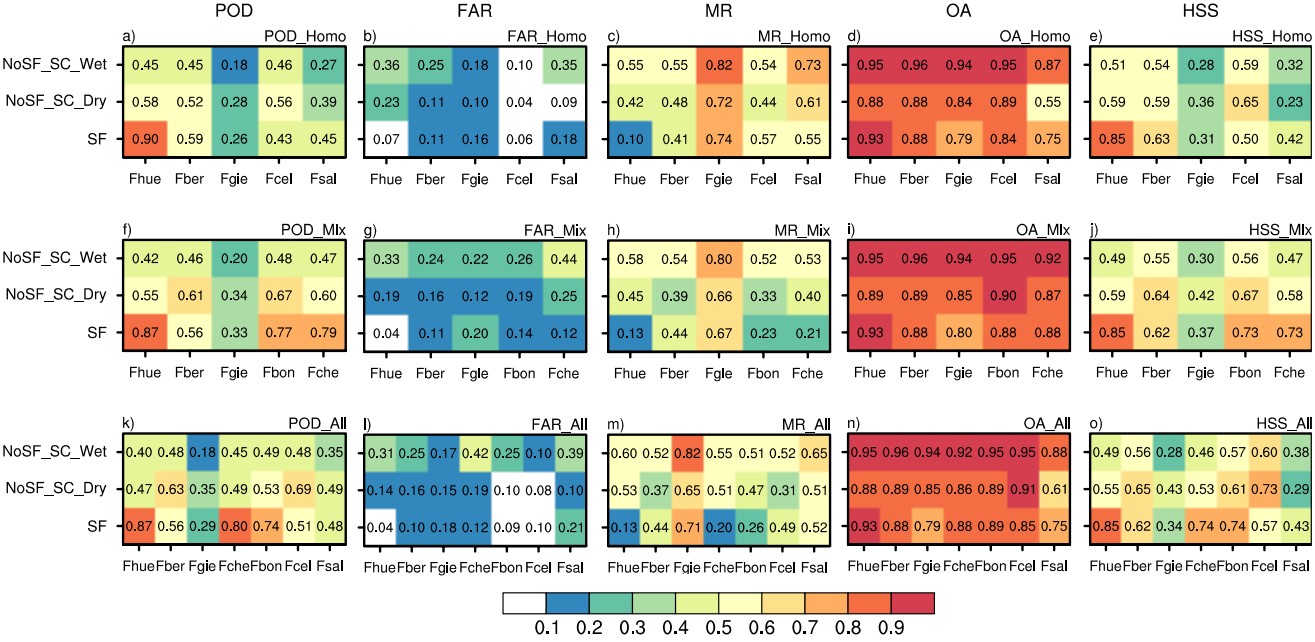

**Figure 10.** The assessment indicators of DTMs trained based on observations from 5 stations with a homogeneous distribution (a-e), inhomogeneous distribution (f-j) of feature variable, and the DTM trained based on observations from all stations.

To explore the accuracy of temporal prediction further, two datasets were created and used in two sensitivity experiments
(the Test 3 listed in Table 2). Both datasets included observations from 5 stations, and the major difference between them was
whether the differences in the distribution of feature variables across stations were significant. One dataset was composed of
observations from Fmor, Fcmb, Fmon, Fsal and Fcel stations (Mix), with substantial differences in the distributions of feature
variables between the first three stations and the last two stations, particularly the WSMAX. The other dataset contained a
more homogeneous distribution of observations compared with the Mix (Homo, observations from Fmor, Fcmb, Fmon, Fbon
and Fche were included). When models were trained with one of the datasets, observations from the stations which were not
included in the training dataset were used to assess its predictive performance. The results of these two sensitivity tests will
also be compared with the performance of models constructed using 80% of observations from all stations to explore the effect
of different numbers of training stations on the model accuracy.

As shown in Figure 10, the model trained with either the Mix (Figure 10i) or Homo (Figure 10d) dataset presented similar
OA with the SIDTM (Figure 8n) when evaluated at the same stations. The accuracies of these models were comparable to that
of the SIDTM, and even to models constructed based only on observations from an individual station. Moreover, the source
of the training samples had a minimal impact on the model performance, as shown in the assessment conducted at Fhue station
(comparing the first and second row of Figure 10). These results indicated that the DTM shows robust spatial transferability,
and importantly, was independent of the source of the training data and the number of stations used. When compared at Fber,
Fgie, Fcel and Fsal stations, the model trained with observations from all stations (third row) outperformed the model trained
with the Homo dataset (second row) in terms of the POD and HSS indices; improvement was also noted in the MR. However,
poorer performance was noted at Fhue, Fbon and Fche stations when the model was trained with observations from all stations
instead of the Homo dataset, as evident form the lower POD and HSS and larger MR (particularly in dry snow cover conditions).
This divergence may partly be explained by the larger threshold maximum wind speed for the occurrence of blowing snow in
models trained with observations from all stations, which prevents the identification of blowing snow events at stations with
relatively high maximum wind speed (see Fhue station in Figure 11: the frequency of blowing events follows WSMAX).
Meanwhile blowing snow is more likely to be predicted at stations with relatively low maximum wind speeds (i.e., the Fgie
and Fber stations), due to the lower threshold maximum wind speed. The occurrence of blowing snow events was

underestimated at the relatively low maximum wind speed stations, (e.g., Fgie station). The accuracy of estimating blowing

snow events increased as the threshold WSMAX decreased.

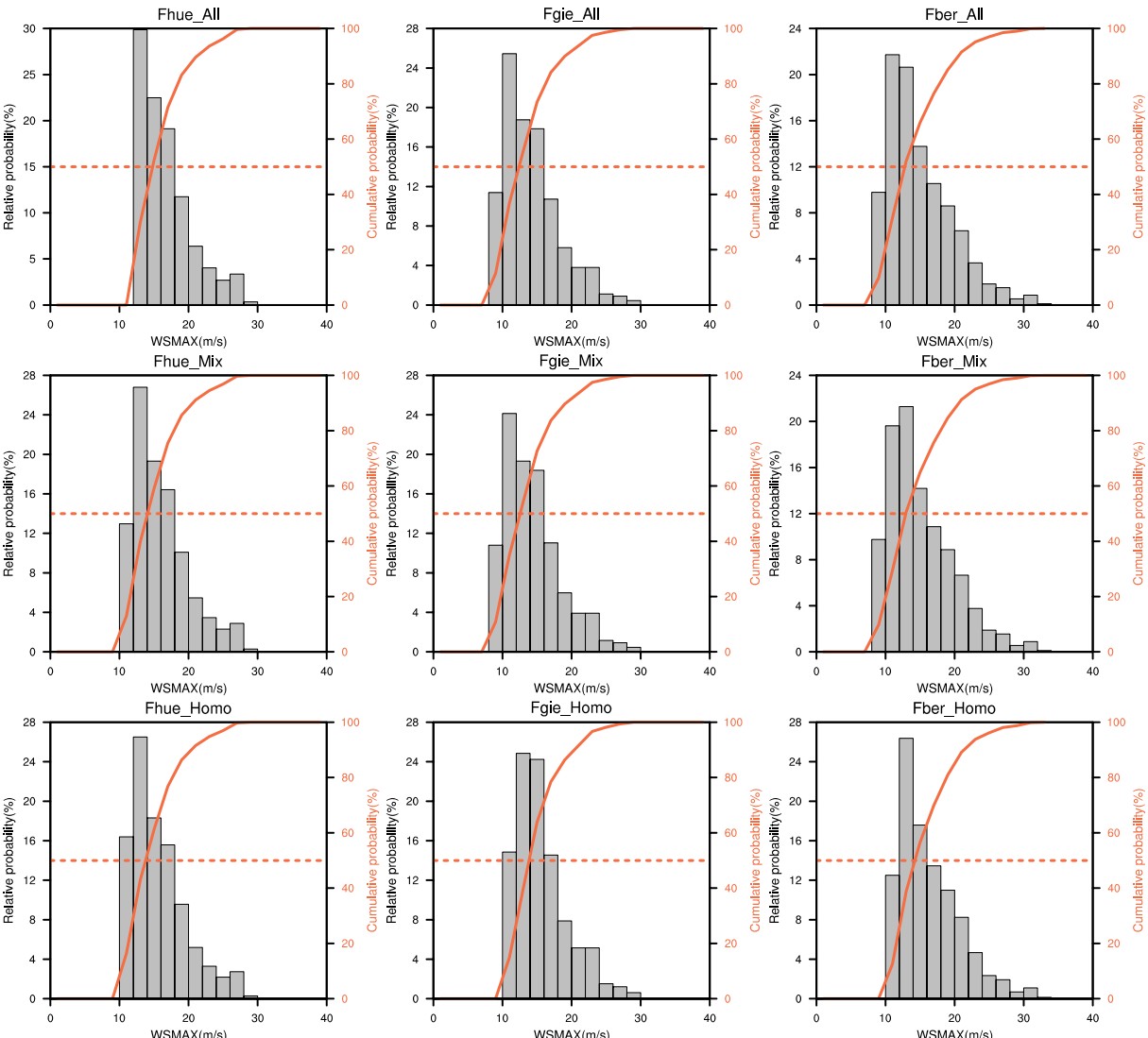

**Figure 11.** The same as Figure 6, but only for the DTM constructed based on observations from all stations (All), and observations from 5 stations with a heterogeneous distribution (Mix) and homogeneous distribution (Homo) of feature variables, respectively. DTMs were assessed at stations with relatively high maximum wind speed (Fhue), and with relatively low maximum wind speed (Fgie and Fber).

The DTMs were established on the principle of maximum inclusiveness, enabling their applicability across broad spatial and temporal scales by synthesizing all the features in the available observations. However, conventional algorithms are often biased towards the majority class, without considering the data distribution. When there is a need to handle heterogeneous data

from various sources, the model must seek a good compromise between accuracy, efficiency and a good fit to all the training samples; therefore, the probability of compromise increases as data heterogeneity increases. This is an important issue to be aware of when using DTMs. In this case, data preprocessing (e.g., the scenario classification method used in this study) is an important and effective step to reduce the heterogeneity of data and to improve the effectiveness of the model.

## 4 Discussion

### 4.1 Influence of training data on the DTM

At stations where a low probability of blowing snow detection was observed (i.e., Fber and Fgie: Figure 5), a considerable improvement in the OA, POD, HSS and MR was achieved when the DTM was trained with the Mix dataset instead of the Homo dataset. For example, a 17.3% increase in the POD, an increase in HSS from 0.59 to 0.64, and a decrease of MR (-0.09) were achieved at Fber station under dry snow cover conditions. Distinct difference was noted at Fber, Fgie, Fcel and Fsal between models trained with or without heterogeneous training samples. However, the source of the training dataset and the distribution characteristics of predictor variables, exerted only a slight impact on the model performance at other stations (e.g., Fhue). In general, using training data with heterogeneous information can effectively improve the classification accuracy while promoting the spatiotemporal transferability of the DTM.

Models did not show consistent improvement in estimating blowing snow occurrence at all stations when trained with 80% of all observations. For example, as mentioned earlier, the model showed a marked decrease of POD and increase of MR in dry snow cover conditions at Fhue, Fche and Fbon stations, revealing that heterogeneity of the training data is a key factor in influencing the performance of the DTM. Meanwhile, the results also highlighted the importance of a reliable and impartial training dataset; strongly imbalanced datasets should be avoided.

### 4.2 Influence of meteorological and environmental factors on the DTM

The occurrence of blowing snow events depends largely on a combination of meteorological and environmental factors, such as wind speed, wind direction, air temperature, topography and vegetation. Wind speed serves as the driving force for the initiation of blowing snow. As it is difficult to determine the occurrence of blowing snow on a deterministic, physical basis (Li and Pomeroy, 1997b), wind speed has been widely used in empirical formulae as a critical indicative parameter to simulate the wind transport of snow (Schmidt, 1980; Pomeroy and Gray 1990). A DTM is a black-box model, in which it is difficult to envisage how the different components operate and interact. However, sensitivity tests have demonstrated that the maximum wind speed played a decisive role in the model predictions. The threshold wind speed for snow transport refers to the minimum wind speed required to initiate the saltation of snow. Once the fastest wind speed exceeds the threshold wind speed, the wind shear stress overcomes the snow cohesion, bonding and frictional resistance, thereby initiating a blowing snow event. The

blowing snow process can then be sustained by a relatively low wind speed. Thus, the maximum wind speed (instead of the mean wind speed) contributes most strongly to the classification accuracy of the DTM.

Air temperature, one of the most critical parameters affecting the microstructural structure and internal physical properties of the snowpack, is associated with snow cohesive resistance. Cohesive resistance increases considerably when snow becomes wet, as water increases the cohesive bonding force between particles. This leads to a sharp contrast of the threshold wind speed for snow transport between wet snow and dry snow. As demonstrated by Li et al. (1997b), condensation and crystal growth occur in the snowpack when the saturation vapor pressure is low, leading to a gradual increase in snow particle bonding resistance and lowering the probability of blowing snow occurrence. Relative humidity, the ratio of vapor pressure and saturation vapor pressure, it therefore has important implications for snow aging processes (i.e., metamorphism, despite Armstrong and Brun (2008) reported that snow metamorphism in alpine snowpack is mainly driven by temperature-gradient). Wind direction is also an important factor influencing blowing snow events, and is closely associated with topography and wind speed (Roebber et al., 2003). Wind speed can vary considerably with wind direction, as site-specific topography may preclude blowing snow under certain wind directions at particular stations. Preliminary studies have suggested an improvement in blowing snow estimates when taking into consideration the wind direction (Baggaley and Hanesiak, 2005; Vionnet et al., 2018). However, for the consideration of the highly site-specific wind speed and large variations of prevailing wind speed across stations, the impact of wind direction is not considered in this study in constructing the DTMs. Generally, snow is eroded from wind-exposed surfaces (e.g., flat surfaces, hilltops, windward slopes, and sparsely vegetated surfaces) and deposited in wind-sheltered areas such as densely vegetated surfaces and topographic depressions (Li and Pomeroy 1997a; Liston and Sturm, 1998, Xie et al., 2019). The topography also very site-specific, and quantifying its potential impacts on blowing snow occurrence is challenging. Thus, the temporal transferability of the DTM is likely to drop sharply and more widespread adoption of the DTM will be hindered once the wind direction and topography are used as feature variables. Vegetation can be effectively quantified by LAI or NDVI; however, the sparse stations in the study region limits its usage here.

## 4.3 Potential sources of error in the DTM

The FlowCapt sensor is sensitive to soil particles, resulting in false alarms for blowing snow events (Vionnet et al., 2018). Therefore, one of the greatest uncertainties is attributed to unreliable blowing snow events recorded by the FlowCapt sensors. Although records corresponding to detected blowing snow events with an absence of simultaneous snow cover and snowfall were removed, unreliable blowing snow events may still exist in the dataset with strict quality control applied. Internal defects of the FlowCapt sensor (in terms of hardware and numerical processing) are another important source of uncertainty (Trouvilliez et al., 2015), although the suitability of the instrument in measuring blowing snow has been evaluated and results have demonstrated its reliability in blowing snow studies (Chritin et al., 1999; Cierco et al., 2007; Das et al., 2012; Trouvilliez et al., 2015). Nevertheless, the rate of snow transported recorded by the FlowCapt can be underestimated (Trouvilliez et al., 2015). Therefore, as the occurrence of blowing snow events was determined based only on the FlowCapt measurements,

inevitable uncertainties exist in this study. To minimize the underestimation of blowing snow measured by the FlowCapt, all records with blowing snow fluxes exceeding 0 were classified as blowing snow event in this study. This is different from Trouvilliez et al. (2015), who used a higher (non-zero) threshold value to remove non-significant blowing snow occurrence when processing FlowCapt measurements.

The problem of strongly skewed data distribution is rather common in real-world applications, and introduces unique challenges when training machine learning models. The term 'imbalanced data' typically refers to the problem where the number of different classes of data is not equally distributed. In this study, for example, the blowing snow events are generally outnumbered by the non-blowing snow events. Learning from imbalanced data has been the subject of many papers, workshops, special sessions and dissertations. However, there is no definite solution. In practice, data imbalance is addressed by a number of methods: using ensemble cross-validation to justify the model robustness; under-sampling the majority class or oversampling the minority class (Zhou and Liu, 2006); or assigning different weights to balance the ratio for each category (Jo and Japkowicz, 2004).

## 4.4 Comparison with other indirect methods

It was extremely difficult to distinguish unreliable records in the quality-controlled data, and it is inevitable that the use of these records in the construction of the DTMs negatively affects the model's skill in detecting blowing snow events. Despite these shortcomings, the calculated accuracies for both the SIDTM and the SSDTMS were superior to (and in most cases much better than) the empirical parameterization schemes using (i) constant threshold wind speed (7.7 m s$^{-1}$ for dry snow transport and 9.9 m s$^{-1}$ for wet snow transport, abbreviating as Constant_dry and Constant_wet, respectively. Li and Pomeroy, 1997a), and (ii) the dynamic threshold adapts to the evolution of air temperature (abbreviating as Ut(10), Ut(10)_wet and Ut(10)_dry shares the same expression but used to detect wet snow and dry snow transport, respectively. Li and Pomeroy, 1997a). As shown in Figure 12, the dynamic threshold wind speed outperforms overwhelmingly the constant threshold wind speed in detecting blowing snow occurrence, while the former's performance evaluation metrics (except for the FAR) characterizing the ability and efficiency in blowing snow detection are inferior to the DTMs obviously (corresponding metrics are shown in Figure 4, 5, 8 and 9, respectively), particularly the POD and HSS. A larger FAR achieved by DTMs indicates a high probability of false blowing snow detection by the data-driven model. When compared to the results from the S2M-Sytron (Vionnet et al., 2018), an avalanche hazard forecast model driven by high spatial and temporal resolution meteorological forcing data downscaled by SAFRAN (Durand et al., 2009), the values for POD and HSS in this study were similar to, or better than, the skill reported for numerical simulations using S2M-Sytron forced with downscaled SAFRAN input. However, due to the limited capacity of DTMs to detect real blowing snow occurrences, particularly under snow cover conditions, large differences exist when compared to R2 (an S2M-Sytron simulation using an updated parameterization for falling snow properties) and R3 (uses the observed 10-m wind speed and direction based on R2). The wide gap between the data-driven model and the physical

constraints model highlights the need to conduct further experimental investigations and analyses to illustrate the limitations of the DTM in detecting blowing snow occurrence. This could shed more light on future developments.

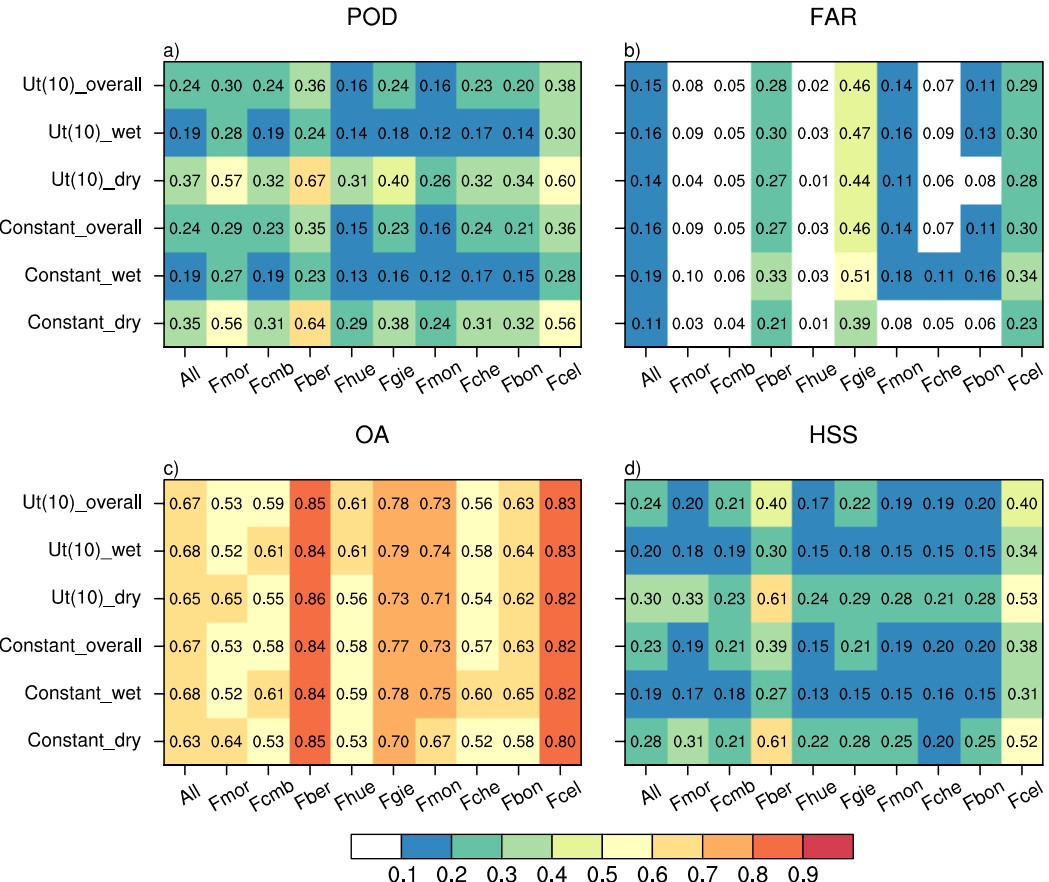

**Figure 12.** The POD (a), FAR (b), OA (c) and HSS (d) of indirect methods using constant threshold wind speed (7.7 m s$^{-1}$ for dry snow transport and 9.9 m s$^{-1}$ for wet snow transport, abbreviating as Constant_dry and Constant_wet, respectively)
and dynamic threshold adapts to the evolution of air temperature (abbreviating as Ut(10), Ut(10)_wet and Ut(10)_dry shares the same expression but used to detect wet snow and dry snow transport, respectively). Constant_overall and Ut(10)_overall is the synthetical metric of wet snow and dry snow conditions.

We note that only those blowing snow events with a snow flux exceeding a threshold of 1 g m$^{-2}$s$^{-1}$ were analyzed, and that if this threshold value is applied in this study, the ability of the DTM to accurately detect blowing snow occurrence is
540 projected to increase. In summary, a simple DTM constructed from conventional meteorological observations is therefore shown to be capable of detecting blowing snow events with a skill superior to the commonly used empirical parameterizations. However, there is still considerable room for improvement when compared with numerical models containing detailed

representations of physical processes, but handling the data imbalance issue appropriately, and minimizing the potential uncertainty resulting from blowing soil particles, are a top priority.

### 545 **4.5 Possible future directions**

Machine learning techniques can be a suitable way to reduce the process complexity and computational cost of traditional physically based blowing snow models. The complex interactive processes of ambient atmospheric conditions make a proper representation of blowing snow processes in the conventional blowing snow model challenging. A major difference between the process-based algorithms and machine learning based models is that the model structure for the former is based on

underlying physical principles, whereas machine learning based model is completely data-driven. Although machine learning models have outperformed simple statistical models and have been widely used in Earth Sciences, they tend to violate physical laws constraints leading to unrealistic predictions (Zhao et al., 2019). Physically-based models, although more complicated, tend to have superior interpretability (Pan et al., 2020). Complementing a physically-based model and machine learning based model is a feasible way for further development. Integrating these two types of models with divergent model structures and

introducing physical constraints on machine learning based models are two popular strategies (Pal and Sharma, 2021). One common way to integrate the constructed DTMs with blowing snow model in detecting blowing snow occurrence is to ensure the application of DTM only when the climatology shares similar distribution property to the training data. In this case, a huge amount of training data with different distribution properties is an essential prerequisite to the performance of DTMs applied over a large area; otherwise, in case of rare events, the detection of blowing snow occurrence relies on the physically-based

blowing snow model. On the other hand, using blowing snow model outputs to pose physical constraints on the DTMs makes the hybrid model capable of learning the nonlinear relations from the training data while obeying the physical laws.

### 5 Conclusions

The accurate classification of blowing snow events is important in numerical models which include blowing snow, as it determines whether or not invoke the parameterizations for sublimation and transport associated with blowing snow processes

in the model (Lenaerts et al., 2012b; Xie et al., 2019). This paper described the construction and evaluation of the machine learning based DTM in detecting blowing snow occurrence in the European Alps. Here, DTMs were trained with routine meteorological observations (WS, WSMAX, T and RH).

An optimal ratio of 0.8 between the training subset and validation subset was chosen here when accuracy, efficiency and reliability of the DTM were taken into consideration. In snowfall conditions, SSDTMs and the SIDTM were trained based on

WSMAX and RH (at stations where RH observations are available), while in snow cover conditions, the models were trained with WS, WSMAX, T and RH (at stations where RH observations are available). Twenty repetitions of a random sub-sampling validation test showed that the maximum wind speed contributes the most to the classification accuracy of the DTMs, and models constructed using additional characteristic attributes achieved higher classification accuracy for blowing snow event

detection. Both the SSDTMs and SIDTM showed strong capabilities for accurately detecting blowing snow; however, notable
variations were seen between stations and conditions. The actual blowing snow events occurring in snowfall conditions were
detected accurately at all stations except for Fber, Fgie and Fcel. However, in non-snowfall conditions, ambient meteorological
conditions exerted complex, nonlinear impacts on the properties and structures of snow particles, hindering the accurate
detection of blowing snow occurrence. The relatively low PODs at Fgie, Fcel and Fber were attributed to the significantly
lower maximum wind speed than that at other stations, which cannot be well captured by the DTM. Overall, 73% and 69% of
blowing snow events occurring under snowfall conditions and dry snow cover conditions were accurately detected by the
SIDTM, but this proportion dropped to 41% for wet snow surfaces.

The constructed DTMs demonstrated good performance in temporal extrapolation, and were also able to accurately detect
blowing snow occurrence at stations outside the training range. The spatial transferability is likely to decline when models are
trained with strongly heterogeneous feature variables. Therefore, in some cases, a few representative predictor variables should
be selected, and data preprocessing (e.g., the scenario classification method used in this study) should be applied to reduce the
heterogeneity of the dataset and improve the effectiveness of the DTM. In summary, both the SSDTMs and SIDTM are useful
tools in detecting the occurrence of blowing snow events, and achieve acceptable accuracy in terms of their spatiotemporal
predictions.

Progress towards the accurate estimation of blowing snow events at local scales relies largely on physically-based blowing
snow models driven by high-resolution meteorological inputs that include a detailed representation of the effects of ambient
atmospheric conditions on the initiation and persistence of blowing snow processes. However, using such models can be
challenging due to the high computation cost of such simulations and the difficulty of obtaining reliable field observations for
the required input. The DTM, constructed from limited available observations, may provide a useful alternative method.
Therefore, DTMs can facilitate research into blowing snow in data-scarce areas such as the Tibetan Plateau, where ten
FlowCapt instruments have been set up and are currently in operation.

*Data availability.* Observation data from the ISAW stations can be accessed at http://www.iav.ch.

*Code availability.* The code to construct a decision tree model in detecting the occurrence of blowing snow is available from
600 GitHub (https://github.com/zpxie-cas/DTM).

*Author contribution.* ZPX, MWQ, YMM and ZYH were responsible for the conceptualization. ZPX, GHS, YZH, WH, RMZS
and YXF were responsible for data searching and processing. ZPX performed the formal analysis and prepared the manuscript
with contributions from all co-authors.

*Competing interests.* The authors declare they have no conflict of interests.

*Acknowledgments.* This work was supported by the National Natural Science Foundation of China (41905012), the Second
Tibetan Plateau Scientific Expedition and Research (STEP) program (2019QZKK0103), and the China Postdoctoral Science
Foundation (2018M641489). We would like to express our special appreciation and thanks to the Scikit-Learn community and
two anonymous reviewers for providing many constructive comments.

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
