# Peer review of "Decision tree-based detection of blowing snow events in the European Alps"

_Hydrology and Earth System Sciences, 2021_

## Author Comment (AC1)

The manuscript described the construction and evaluation of the machine learning based decision tree model (DTM) in detecting blowing snow occurrence by exploiting routine meteorological observations from 10 sites in the European Alps. The constructed DTMs demonstrated good performance in temporal extrapolation, and were also able to accurately detect blowing snow at stations outside the training range. In general, both the SSDTMs and SIDTM are useful tools in detecting the occurrence of blowing snow events, and achieve acceptable accuracy in terms of their spatiotemporal predictions.

Traditional blowing snow occurrence or simulation in land surface model or climate models has the difficulty of obtaining reliable observations for the required input. However, this study developed a simple but efficient tool to detect blowing snow occurrences and to advance our understanding of the relationships between blowing snow processes and ambient meteorological conditions. It also provide a potential insight to the future machine learning studies and modeling studies for the blowing snow events in the highland areas, such as the Tibetan Plateau.

Scientifically, the machine learning based DTM and the evaluations are well conducted and the results are reasonable. The manuscript of this paper is well organized and it is logical in its presentation. I believe this manuscript is suitable for publication in HESS, but have some primary concerns shown as following. I suggest publication of this paper with following minor revisions.

**Response**: We thank the reviewer very much for their constructive comments on our manuscript. We have studied those comments carefully and have made corrections accordingly which we hope meet with approval. The main changes in the manuscript, in response to the reviewer's suggestions, are the following (changes in the manuscript are in blue):

1. Could you give a schematic flowchart of the machine learning based decision tree model (DTM)? This would enrich your manuscript and also make it easier for the reader to understand.

   **Response:** The following schematic flowcharts have been added to the manuscript. The first one is used to clarify the procedures to identify the presence of snow on the ground, while the second one is to show the logical framework of the work and in the second one, we choose WSMAX and T as the two key feature variables to construct the simplest DTM to detect blowing snow occurrences.

[Figure]

Figure 1. Schematic flowchart of a) the procedures to identify the presence of snow, and b) flowchart of a simple decision tree model to detect blowing snow occurrence (only WSMAX and T were used to construct the DTM, A denotes the threshold maximum wind speed, B1 and B2 denote the threshold air temperature), and c) logical framework of this study.

2. How is the DTM's computation cost? Your study just involve 10 sites over a small region. I'm not sure how much computation time it would take if this method was applied to large area or other regions.

Response: Special thanks to you for your concerns to the DTM's computational cost. Actually, the computation cost of constructing and assessing the DTM is pretty low. For example, when constructing the site-independent decision tree model (SIDTM) in condition which surface covered by dry snow and without concurrent snowfall with observations from all stations (327387 samples in total, with WS, WSMAX, T and RH as feature variables), the total execution time is 1.6s. What's more, to reduce the classification uncertainty attributable to training data selection, twenty repetitions of a random sub-sampling validation method were applied in the construction of each decision tree model, and the decision tree model constructed under this condition is the most complex and the highest computational cost in this study. Therefore, there is no need to worry about the computational cost when apply this method to large area or other regions, the high computationally-efficient of the decision tree model has been well tested in this study.

3. At line 38, it would be better to replace the "causing a loss of visual sight" by "causing severe reductions to visibility near the ground".

Response: We have made correction according to the Reviewer's comment.

4. Replacing the word "distribution" in line 40 by "redistribution".
   **Response:** Correction has been made.

5. The definite article "the" in the "at the local scale" in line 47 is needless.
   **Response:** We agree and have deleted the needless word.

6. At line 51, some references should be added to describe alternative methods that have been proposed.
   **Response:** Thanks for pointing this out. We have added the following references:
   [1] He, S. W., and Ohara, N.: A New Formula for Estimating the Threshold Wind Speed for Snow Movement. Journal of Advances in Modeling Earth Systems, 9(7), 2514-2525, 2017.
   [2] Li, L., and Pomeroy, J. W.: Estimates of threshold wind speeds for snow transport using meteorological data. Journal of Applied Meteorology, 36(3), 205-213, 1997a.
   [3] Schmidt, R. A.: Threshold Wind-Speeds and Elastic Impact in Snow Transport. Journal of Glaciology, 26(94), 453-467, 1980.

   Alternative methods using empirical formulae to parameterize blowing snow occurrence have been proposed (e.g., He and Ohara, 2017; Li and Pomeroy, 1997a; Schmidt, 1980).

7. It would be better to replace "this method" in line 71 with "the remote retrieval algorithms".
   **Response:** Correction has been made. Thanks.

8. Maybe you need to check the manuscript again. For example, at line 504, "…….., and can were also able to accurately……" It seems the "can" is needless.
   **Response:** Agreed. A thorough check was made to the manuscript to improve the accuracy of language.

9. Please give the full name of ISAW while it first appears in the article.
   **Response:** The full name of ISAW is not available because the website only provides the abbreviated station name. Other studies (e.g., He and Ohara, 2017; Vionnet et al., 2018) use the abbreviated station name as well. We hope you will understand that.
   [1] He, S. W., and Ohara, N.: A New Formula for Estimating the Threshold Wind Speed for Snow Movement. Journal of Advances in Modeling Earth Systems, 9(7), 2514-2525, 2017.
   [2] Vionnet, V., Guyomarc'h, G., Lafaysse, M., Naaim-Bouvet, F., Giraud, G., and Deliot, Y.: Operational implementation and evaluation of a blowing snow scheme for avalanche hazard forecasting. Cold Regions Science and Technology, 147, 1-10, 2018.

---

## Author Comment (AC2)

This paper presents the development a new method to detect the occurrence of blowing snow event in alpine terrain. The authors built decision tree classification models (DTMs) using data from automatic stations measuring simultaneously standard meteorological observations and blowing snow fluxes. They proposed to develop decision trees purely based on standard meteorological observations. The authors found that maximal wind speed had the largest influence in determining the occurrence of blowing snow events for different conditions. They also illustrate how the selection of the stations in the training set influence the performances of the resulting DTMs.

The subject of this paper is interesting for a large community studying snow in mountainous region for various applications: avalanche hazard forecasting, mountain hydrology, road maintenance, … The estimation of blowing snow occurrence is also a key in blowing snow models. My main comment about the study concerns the absence of quantification of the benefit of using DTMs compared to more traditional methods to predict blowing snow occurrence. Machine learning is still new in the community and it would be very valuable if the authors could quantify it benefit compare to more traditional methods. This question needs to be clarified prior to publication in HESS. It is detailed below as a general comment followed by more specific and technical comments.

**Response:** We thank the reviewer very much for their constructive comments on our manuscript. Comparison of the DTM and traditional methods (using constant threshold wind speed and dynamic threshold wind speed) have been made to quantify their ability in detecting blowing snow occurrence. We hope this can release the concerns that you have. For detailed descriptions please refer to the response to the Comment 25.

**General Comment**
Different methods have proposed to obtain an estimation of the threshold wind speed for snow transport. When combined with information on wind speed, theses methods can be used to estimate the occurrence of blowing snow events. The simplest method consists in constant threshold value for dry and wet snow (see for example the average values reported on page 209, section 4 of Li and Pomeroy, 1997). In the same paper, Li and Pomeroy (1997) proposed a slightly more complex formulation of the threshold wind speed that only depends on air temperature. He and Ohara (2017) proposed a more advanced formulation that depends on snow particle size, deposition time and temperature. Despite being more complex, this formulation can still be derived using standard meteorological measurements (precipitation, temperature). Finally, formulations of the threshold wind speed that depend on the physical properties of the snowpack have been developed. Liston et al. (2007) used a formulation that depends on the surface snow density whereas Schmidt (1980), Guyomarc'h and Merindol (1998) et Lehning et al (2000) proposed formulation that depends on the microstructure of the surface snow.

The decision trees developed in this study are a new method proposed to determine the occurrence of blowing snow. For the community studying snow in mountainous region, the authors should document the performance of this new approach compared to previous approaches used as benchmarks. So far, it is very difficult for the reader to estimate the benefit using decisions-trees compared to more standard methods. The

significance of the paper would be certainly improved it the authors can show that their DTMs can provide improved the detection of blowing snow occurrence compared to more standard methods.

All the formulations of the threshold wind speed depending on physical properties of surface snow cannot be tested here since they require a snowpack model. However, the decision trees could still be compared with the performances obtained for three formulation the threshold wind speed of various complexity: (i) constant threshold for dry and wet snow, (ii) Li and Pomeroy (1997), (iii) He and Ohara (2017). These threshold wind speed could be computed at an hourly time step at all the ISAW stations and compared with mean wind speed and/or maximum wind speed to determine blowing snow occurrence. OA, POD, FAR and HSS could be then computed and compared with the results of the DTMs. Such evaluation would highlight the benefits and limitations of the decision trees and would guide future developments.

**Response:** We appreciate the time and effort you have dedicated to providing insightful suggestions to strengthen our paper. We help that our modifications and the responses we provide below satisfactorily address all the issues and concerns you have noted (The main changes in the manuscript are in blue).

The comparisons have been added in the reversed manuscript based on your suggestion. However, we only compared the performance of the methods using constant threshold wind speed and dynamic threshold wind speed proposed by Li and Pomeroy (1997) with DTM. More specific interpretation is given in the response to the Comments 25.

**Specific comments**

1. Abstract L 21-22: it would be interesting to clarify in the abstract that snow depth and measurements are used as well to determine the occurrence of blowing snow. Indeed, without snow on the ground, blowing snow cannot occur.
   **Response:** We agree with you and have incorporated this suggestion throughout our manuscript.
   The DTMs were constructed based on routine meteorological observations (mean wind speed, maximum wind speed, air temperature and relative humidity) and snow depth measurements (including in-situ snow depth observations and satellite-derived products).

2. Abstract L 25: the authors should explain which atmospheric variables have a divergent distribution.
   **Response:** Yes, the following change has been made. Thanks a lot.
   However, the spatiotemporal transferability of the DTM might be limited if the divergent distribution of wind speed exists between stations.

3. P2 L 48-49: I recommend the authors to also include the original references for the different sensors measuring blowing snow fluxes:
   SPC: Sato, T., Kimura, T., Ishimaru, T., & Maruyama, T. (1993). Field test of a new snow-particle counter (SPC) system. Annals of Glaciology, 18, 149-154.

FlowCapt: Chritin, V., Bolognesi, R., & Gubler, H. (1999). FlowCapt: a new acoustic sensor to measure snowdrift and wind velocity for avalanche forecasting. Cold Regions Science and Technology, 30(1-3), 125-133.

**Response:** We have included these references you recommended.

For example, the mechanical traps used by Budd et al. (1966), the optical sensors deployed in the Antarctic and Alps (Nishimura and Nemoto, 2005; Vionnet et al., 2013), and the acoustic sensors (i.e., FlowCapt and SPC) used to provide reliable measurements of blowing snow mass flux (Chritin et al., 1999; Sato et al., 1993; Trouvilliez et al., 2015).

4.  P2 L 53-57: The large increase in surface snow cohesion in presence of liquid water (resulting from rainfall or melting) is not the only factor affecting the evolution of threshold wind speed contrary to what is suggested here by the authors. I recommend to explicitly mention the impact of sintering od dry snow (Schmidt, 1980, He and Ohara, 2017). In addition, mechanical fragmentation of snow grains during blowing snow events (Comola et al., 2017) can favor snow sintering and affect the evolution of the threshold velocity (Vionnet et al., 2013).

    **Response:** Thank you very much for pointing this problem out. In order to maintain the highest possible standard of rigorous precision in the description, we have made the following modification based on your suggestion.

    Previous studies have demonstrated that cohesive resistance increases dramatically when snow becomes wet, as the melt water increases the associated cohesion between the particles (e.g., Li and Pomeroy, 1997; Schmidt, 1980), and sintering of snow particles have a significant bearing on the cohesive force development as well (He and Ohara, 2017; Schmidt, 1980).

5.  P 2L 57-58: the dependence of the threshold wind speed on air temperature and humidity is only indirect and is due to the influence of air temperature and humidity on the cohesion of surface snow. It would be interesting to explicitly mention it in the introduction so that it is clear for the reader.

    **Response:** Yes, we agree with you and the following change has been made to clearly show the relationships between air temperature, humidity and the threshold wind speed.

    As summarized by Schmidt (1980), the threshold wind speed highly depends on the cohesion between snow particles and was greatly influenced by temperature, humidity and deposition time.

6.  P 2 L 60: Liston et al (2007) are only reporting values from previous studies for the threshold wind speed for fresh snow. I recommend the authors to refer to papers that measured the threshold wind speed for snow transport in the field: Li and Pomeroy (1997), Guyomarc'h and Merindol (1998), Doorschot et al (2004), Clifton et al. (2006).

    **Response:** We have incorporated your recommendation by revising the following content.

Threshold wind speed was found to be 9.9 m/s for wet snow and 7.7 m/s for dry snow, and a formula expresses the threshold wind speed as a function of air temperature has been proposed based on field observations from the Canadian Prairies (Li and Pomeroy, 1997a).

7. P 3 L 90-91: I think the author should add snow depth in the list of variables because they are still using this measurement to determine if there is snow on the ground or not. If snow depth is not used at all, I would find very distributing to have an algorithm that predict blowing snow occurrence without even checking if there is presence of snow on the ground.

   **Response:** Indeed, snow depth measurement is used to determine the presence of snow on the ground. Based on your comment, snow depth has been added to the list of variables used in this study.

   In this study, we use a machine learning based decision tree model (DTM) to detect the presence of blowing snow by exploiting routine meteorological observations (such as wind speed, wind direction, air temperature, precipitation and relative humidity) and snow depth measurements from 10 ISAW stations (http://isaw.ch/).

8. P 4 L 97: does the reported height (3.5 m) correspond to the height of the wind sensor about snow-free ground? If it is the case, are the authors using the snow depth measurement to adjust the wind speed at a constant height during the course of the winter? This may have a slight influence if enough snow is accumulated below the wind sensor. See for example Vionnet et al. (2013).

   **Response:** Yes, we had noted the influence of accumulated snow on the wind speed. However, observed snow depth fluctuated wildly in most stations, and there was a large difference in the measured snow depth from two nearby mounted snow depth sensors. Therefore, the snow depth measurements were only used to determine the presence of snow on the ground, rather than adjusting the wind speed to a different but constant height. A rise in the uncertainty of the adjusted wind speed seems inevitable once the influence of accumulated snow was not taken into account, but we think this compromise is acceptable because the large difference of the two snow depth measurements may introduce larger uncertainty in the wind speed adjustment. We hope that our response would address your concerns.

9. P 5 L 107-115: It would be interesting to describe in this paragraph how the authors used the signal of the snow depth (SD) sensor to identify the periods when the surface was covered by snow. Did they use SD> 0 and another threshold value?

   **Response:** The observed snow depth from the nearby mounted snow depth sensors were used together with the CryoLand fractional snow cover product over Alps (with daily temporal frequency and 250 m spatial resolution) and the MODIS snow cover product (MOD10A1 and MYD10A1, with daily temporal frequency and 500 m spatial resolution) to determine the presence of snow on the ground. As the MODIS instrument provides high spatial resolution views of each point on the earth four times per day, we think the MODIS can provide more accurate snow cover

information than the CyroLand product at an hourly time scale. Therefore, once whether the surface was covered by snow or not cannot directly be determined by snow depth measurements (signals of the two snow depth sensors were lost or large difference exist in the two snow depth measurements), the presence of snow on the ground is determined based on the most adjacent MODIS observation. If the most adjacent MODIS product shows that the station is covered by cloud or the data quality is bad, the second most adjacent MODIS observation is used. If all these two adjacent MODIS observations are unavailable, the CyroLand product is used.

Meanwhile, a schematic flowchart was added to clarify the procedures to identify the presence of snow, and the following content was included in the revised manuscript. The presence of snow on the ground was determined based on the snow depth measurements from two snow depth sensors, the MODIS daily snow cover product (MOD10A1 and MYD10A1, Hall and Riggs, 2021a, b) and the CryoLand fractional snow cover product over Alps (http://cryoland.enveo.at). For detailed procedures please refer to the schematic flowchart in Fig 1a.

[Figure]

10. P 5 L 109: which threshold check was applied to relative humidity?
**Response:** More precisely, we applied main change range check to the relative humidity to processing the raw observations (50% change of relative humidity within an hour was used as the threshold). Modification has been made to make the sentence more efficient and accurate.
First, using a threshold value of 50% change within an hour, the main change range check was applied to the relative humidity to detect its abnormal change. In addition, a threshold check was performed for the hourly measured air temperature, wind speed, maximum wind speed.

11. P 5 L 114-115: the author should explain in this section how they define a blowing snow occurrence. So far it is done in the discussion (L 456-457). All blowing snow fluxes strictly positive are considered so far. Trouvilliez et al. (2015) used a threshold of 1 g/m2/s to remove non-significant blowing snow occurrence when processing FlowCapt measurements. Can the authors comment on the sensitivity of their results to the value of this threshold? Such data processing could potentially improve the ability of the decision trees to detect the main blowing snow events.

**Response:** Thanks for your suggestion, the following content has been added to define a blowing snow occurrence.

In this study, periods of blowing snow occurrence were identified when positive blowing snow flux was observed. This is different from the work of Trouvilliez et al. (2015) who used a threshold of 1g $m^{-2}$ $s^{-1}$ to remove non-significant blowing snow occurrences and the work of Vionnet et al. (2013) who only analyzed events of duration longer than 4 hours.

Based on the preliminary results of our current work, the observed frequency of blowing snow occurrence is sensitive to the threshold. The higher the threshold is, the relatively lower the frequency of blowing snow occurrence. Although the use of a lowered threshold does not affect the derived frequency significantly, the frequency highly depends on the relatively small magnitude blowing snow flux. If we apply a relatively high threshold to select significant blowing snow occurrence for analyzing, the samples with the occurrence of blowing snow events will undoubtedly decrease, resulting in a more serious data-imbalance issue and may have a profound impact on the accuracy and efficiency of the decision tree model. As far as we're concerned, it's hard to say that apply a threshold value could improve the ability of the decision tree model in detecting blowing snow occurrence, because although the significant blowing snow event may easier be accurately captured by the decision tree model, the seriously imbalanced training dataset cannot provide enough useful information to the model to learn to make a correct classification.

12. P5 L 121-123: Compaction due to overburden is not affecting the properties of surface snow so that it is not impacting directly the evolution of the threshold wind speed. The term "viscous" is not really appropriate here and I recommend the author to remove from the text: "and are generally drier and less viscous than the deposited snow".
    **Response:** Yes, we agree with you after carefully think about this description, and based on your suggestion, this content has been removed from the revised manuscript. Thanks.

13. P 5 L 125: was it challenging for the authors to identify blowing snow events with concurrent snowfall? Indeed, the precipitation gauges installed at the ISAW stations must certainly suffer from large wind-undercatch during strong wind events which may potentially lead to no solid precipitation accumulating in the gauge.
    **Response:** Yes, indeed. Because of the impact of strong wind speed on the solid precipitation observation, blowing snow events with concurrent snowfall may be misclassified as NoSF_SC_DRY, and might potentially limit the decision tree model's ability in accurately detecting blowing snow occurrence in NoSF_SC_DRY condition. However, for the moment, blowing snow events with concurrent snowfall can only be effectively identified based on the precipitation observations.

14. P 5 L 130: Can the author explain how they identify the atmospheric conditions for melting snow? Did the authors only consider one hour of positive air temperature to identify melting snow?

**Response:** In this work, dry snow is defined as snow that has not received temperatures of 0 °C or above, such data processing is referred to the practice of Li and Pomeroy (1997). To clarify this issue, the definition of dry snow has been changed with the following text.

while dry snow defines as snow that has not received temperatures of 0 °C or above, or liquid precipitation (Li and Pomeroy, 1997a).

15. P 5 L 130-132: the authors identified 3 types of atmospheric conditions when developing their algorithm. For a better understanding of the importance of these conditions, it would be really interesting if the authors could produce a graph or table that shows the frequency of occurrence of the 3 types at each station and at all stations combined. This graph or table could also show the frequency of blowing snow occurrence for each of the 3 types of atmospheric conditions at each station and at all stations combined. Such graph would highlight that blowing snow events are rare events (as mentioned by the authors at L 335-336). This is important to understand the evaluation metrics derived from the contingency tables.

**Response:** We have incorporated this suggestion by adding the following graph into the revised manuscript.

[Figure]

Figure 2. The frequencies of occurrence of the 3 types of blowing snow at each station and at all stations combined. SF denotes snowfall condition and NoSF_SC_DRY denotes surface covered by dry snow without concurrent snowfall condition, and NoSF_SC_WET denotes surface covered by wet snow without concurrent snowfall condition.

16. P 8 L 197: Can the authors explain the definition of the mean and standard deviation in the context of the evaluation of the DTM? At the moment, it is not clear for a reader that is not familiar with DTM.

**Response:** In this study, to reduce the classification uncertainty attributable to training data selection, twenty repetitions of a random sub-sampling validation method were applied in the construction of each decision tree model. It is necessary to assess the robustness of the decision tree model. Therefore, the variation range and standard deviation of model accuracy were calculated to show the difference between the 20 testing probabilities, and the maximum and minimum overall accuracy were also provided to show the model's ability in detecting blowing snow occurrence under 3 types of atmospheric conditions.

The following modifications have been made:

The variation range and its standard deviation of overall accuracy changed slightly with decreasing sample size: the accuracy range ranged from $1.06 \times 10^{-3}$ to $7.19 \times 10^{-3}$ and the standard deviation increased from $2.23 \times 10^{-4}$ to $15.26 \times 10^{-4}$ as sample size decreased.

17. P 8 L 200: It the selected value of 0.8 is a classic ratio used when training and evaluating DTM? A comparison with a few references from the literature would be certainly appropriate here.

    **Response:** Actually, the split ratio for the training and validation dataset varies considerably in different studies, but its effect on the performance of the decision tree model is not significant. The best practice is to keep the test data set small as compared to the train data set depending on the size of the data set. But the test should not be so small that it is not representative of the samples. 0.5-0.9 was generally used and the 0.7 and 0.8 split ratios performed better as demonstrated by Rácz et al (2021). The training set proportion of 0.8 was used in this study for the consideration of the reliability of the accuracy assessment decreased with decreasing validation sample size.

    Rácz A, Bajusz D, Héberger K. Effect of Dataset Size and Train/Test Split Ratios in QSAR/QSPR Multiclass Classification[J]. Molecules, 2021, 26(4): 1111.

18. P 10 L 230-245: the tests using only single attribute such as air temperature or relative humidity do not make a lot of sense from a physical point of view. Based on the existing and extensive literature on blowing snow, it is well established that wind speed is the primary driver and a necessary condition of blowing snow occurrence. It would be interesting for the reader to present step-by-step and logical construction and tests of the DTMs: (i) tests of the mean wind speed (ii) tests of the benefit of using maximum wind speed since gust affects wind-induced snow transport (Naaim Bouvet et al., 2011), (iii) tests of using additional variables such as air temperature and relative humidity to account for their indirect effect on surface snow cohesion. Such reduced and logical set of tests would also make Figure 1 easier to read.

    **Response:** After careful consideration of this comment posted, we have subdivided the section 3.2 into two parts: 3.2.1 Air temperature and relative humidity and 3.2.2 Mean and maximum wind speed, to present the sensitivity of DTM to the feature variables step-by-step. And we think this adjustment can illustrate the logical set of

tests of the DTM as well. The majority of the content in this section has been adjusted or modified.

**3.2.1 Air temperature and relative humidity**

Of all the attribute combinations evaluated, models trained merely with either T or RH presented the lowest accuracy (Figure 3a-c), indicating that the use of T or RH alone cannot fully capture the variance in the validation samples. However, significant improvements were achieved when either WS or WSMAX were accompanied by T or RH, even though the single factors performed poorly when used alone. Taking Fcmb station in snowfall conditions as an example, model accuracy increased from 0.54 when the model trained merely with T to 0.8 or 0.82 when WS or WSMAX was added, respectively. These results suggest that neither T nor RH is the guarantee of model accuracy, although the model with more predictor variables used generally achieved relatively high accuracy.

**3.2.2 Mean and maximum wind speed**

Models trained with a combination that included WAMAX outperformed the other models, revealing that WSMAX rather than WS contributed the most to the model accuracy, highlighting the importance of WSMAX in constructing a reliable DTM. The result is reasonable, as the fastest wind speed acts as the primary driving force that allows wind shear stress to overcome snow cohesion, bonding and frictional resistance (He and Ohara, 2017). Wind transport of snow can be initiated once the fastest wind speed exceeds the threshold wind speed, and the blowing snow process can then be sustained by relatively low wind speeds. In other words, the fastest wind speed and the mean wind speed control the occurrence and persistence of blowing snow events, respectively. Generally, model accuracy improved as more predictor variables were used. However, strongly correlated feature variables might slightly affect the model accuracy; this was evident when WS was added to WSMAX in snowfall conditions. Overall, this comparison indicates the superiority of DTM as a means of blowing snow identification, which is achieved by making full use of all available feature variables.

19. P 10 L 236-239: it would be interesting to add a few references from the literature to support this description.
   **Response:** Reference has been added.

20. P 14 Figure 5: The 'MR' metric is not described in this paper. (see also Fig. 8, and the rest of the text).
   **Response:** Thank you for pointing out this mistake. The formula for calculating the 'MR' metric has been added.

21. P 17 L 364: Is it only WSMAX which is considered here or the distributions of other features were also considered?
   **Response:** Large difference exists in both the distribution of WS and WSMAX, but only WSMAX is considered here because the snow movement is controlled by the WSMAX.

22. P 20 L 401: it would be interesting if the authors could discuss in this section how these decision trees could be used in blowing snow models. Indeed, as explained at L 265-266, the authors used different decisions trees as a function of the atmospheric conditions. The authors should explain how such concept could be applied in a numerical model.

**Response:** This is a very enlightening suggestion. We have incorporated your comments by adding a new section (4.5 Possible future directions) to suggest future directions for further improvements. The new additions are as follows.

**4.5 Possible future directions**

Machine learning techniques can be a suitable way to reduce the process complexity and computational cost of traditional physically based blowing snow models. The complex interactive processes of ambient atmospheric conditions make a proper representation of blowing snow processes in the conventional blowing snow model challenging. A major difference between the process-based algorithms and machine learning based models is that the model structure for the former is based on underlying physical principles, whereas machine learning based model is completely data-driven. Although machine learning models have outperformed simple statistical models and have been widely used in Earth Sciences, they tend to violate physical laws constraints leading to unrealistic predictions (Zhao et al., 2019). Physically-based models, although more complicated, tend to have superior interpretability (Pan et al., 2020). Complementing a physically-based model and machine learning based model is a feasible way for further development. Integrating these two types of models with divergent model structures and introducing physical constraints on machine learning based models are two popular strategies (Pal and Sharma, 2021). One common way to integrate the constructed DTMs with blowing snow model in detecting blowing snow occurrence is to ensure the application of DTM only when the climatology shares similar distribution property to the training data. In this case, a huge amount of training data with different distribution properties is an essential prerequisite to the performance of DTMs applied over a large area; otherwise, in case of rare events, the detection of blowing snow occurrence relies on the physically-based blowing snow model. On the other hand, using blowing snow model outputs to pose physical constraints on the DTMs makes the hybrid model capable of learning the nonlinear relations from the training data while obeying the physical laws.

23. P 21 L 432: In alpine snowpack, snow metamorphism is mainly driven by temperature-gradient (Armstrong and Brun, 2008). This sentence should be reformulated.

**Response:** This sentence has been replaced by the following content.

Relative humidity, the ratio of vapor pressure and saturation vapor pressure, it therefore has important implications for snow aging processes (i.e., metamorphism, despite Armstrong and Brun (2008) reported that snow metamorphism in the alpine snowpack is mainly driven by temperature-gradient).

24. L 21 L 438: In Vionnet et al. (2018), the wind direction is used to identify the windward side among the virtual slopes surrounding the stations. On the windward slope, blowing snow occurrence is then computed by combining information on wind speed and information on simulated surface snow following the method proposed by Guyomarc'h and Merindol (1998). The wind direction is not used directly as an input variable of the algorithm used to detect blowing snow occurrence. I recommend the authors to modify the text.

**Response:** The following text has been added to clarify that the wind direction is not used in constructing the decision tree model.

However, for the consideration of the highly site-specific wind speed and large variations of prevailing wind speed across stations, the impact of wind direction is not considered in this study in constructing the DTMs.

25. P 22 L 469-480: in this section of the discussion, the authors compare the performance of their decision trees with the performances of a modelling chain for avalanche hazard forecasting that simulates blowing snow occurrence (Vionnet et al., 2018). This modelling chain has been evaluated at the same stations as those used to develop the decision trees. The authors should compare the performance of their decision trees with the simulation R3 used in Vionnet et al. (2018) that used the observed wind speed at the stations to drive the modelling chain. Instead, simulations R1 and R2 from Vionnet et al. (2018) and mentioned by the authors used the wind speed from the SAFRAN analysis that adds uncertainty in the simulated blowing snow occurrence. The fact that the DTMs used in this study achieved similar results to experiment R1 in Vionnet et al. (2018) cannot be used to support the claim made by the authors at L 480-482. As explained in my general comment, the best method to support such claim would be to carry out a systematic benchmark versus standard methods available in the literature.

**Response:** It must be admitted that this comment is pretty valuable and we struggled to address the concern you're mentioned in reviewing the original version of this manuscript submitted to the "Water Resources Research", but failed. The following modifications have been made after we reexamined this suggestion. However, we only compared the performance of the methods using constant threshold wind speed and dynamic threshold wind speed proposed by Li and Pomeroy (1997) with DTM, because the deposition time since last snowfall is a vital parameter in the formula proposed by He and Ohara (2017). Unfortunately, determining its value is impractical because lots of records were discarded after the strict data selection criteria we applied. Besides, direct snow depth observations were considered unreliable because there was large difference in the measured snow depth from two nearby mounted snow depth sensors. But, very comprehensive comparison of the DTM and traditional methods with varying degrees of complexity is highly needed, and this is the ongoing work being carried out. Meanwhile, the content at L 480-482 has been adjusted.

In summary, a simple DTM constructed from conventional meteorological observations is therefore shown to be capable of detecting blowing snow events with a skill superior to the commonly used empirical parameterizations. However, there is

still considerable room for improvement when compared with numerical models containing detailed representations of physical processes, but handling the data imbalance issue appropriately, and minimizing the potential uncertainty resulting from blowing soil particles, are a top priority.

It was extremely difficult to distinguish unreliable records in the quality-controlled data, and it is inevitable that the use of these records in the construction of the DTMs negatively affects the model's skill in detecting blowing snow events. Despite these shortcomings, the calculated accuracies for both the SIDTM and the SSDTMS were superior to (and in most cases much better than) the empirical parameterization schemes using (i) constant threshold wind speed (7.7 m/s for dry snow transport and 9.9 m/s for wet snow transport, abbreviating as Constant_dry and Constant_wet, respectively. Li and Pomeroy, 1997a), and (ii) the dynamic threshold adapts to the evolution of air temperature (abbreviating as Ut(10), Ut(10)_wet and Ut(10)_dry shares the same expression but used to detect wet snow and dry snow transport, respectively. Li and Pomeroy, 1997a). As shown in Figure 12, the dynamic threshold wind speed outperforms overwhelmingly the constant threshold wind speed in detecting blowing snow occurrence, while the former's performance evaluation metrics (except for the FAR) characterizing the ability and efficiency in blowing snow detection are inferior to the DTMs obviously (corresponding metrics are shown in Figure 4, 5, 8 and 9, respectively), particularly the POD and HSS. A larger FAR achieved by DTMs indicates a high probability of false blowing snow detection by the data-driven model. When compared to the results from the S2M-Sytron (Vionnet et al., 2018), an avalanche hazard forecast model driven by high spatial and temporal resolution meteorological forcing data with downscaled by SAFRAN (Durand et al., 2009), the values for POD and HSS in this study were similar to, or better than, the skill reported for numerical simulations using S2M-Sytron forced with downscaled SAFRAN input. However, due to the limited capacity of DTMs to detect real blowing snow occurrences, particularly under snow cover conditions, large differences exist when compared to R2 (an S2M-Sytron simulation using an updated parameterization for falling snow properties) and R3 (uses the observed 10-m wind speed and direction based on R2). The wide gap between the data-driven model and physical constraints model highlights the need to conduct further experimental investigations and analyses to illustrate the limitations of the DTM in detecting blowing snow occurrence. This could shed more light on future developments.

[Figure]

Figure 12. The POD (a), FAR (b), OA (c) and HSS (d) of indirect methods using constant threshold wind speed (7.7 m/s for dry snow transport and 9.9 m/s for wet snow transport, abbreviating as Constant_dry and Constant_wet, respectively) and dynamic threshold adapts to the evolution of air temperature (abbreviating as Ut(10), Ut(10)_wet and Ut(10)_dry shares the same expression but used to detect wet snow and dry snow transport, respectively). Constant_overall and Ut(10)_overall is the synthetical metric of wet snow and dry snow conditions.

26. P 23 L 520: can the authors comment about the availability of the code of the decision trees proposed in this study?

    **Response:** The availability of the code has been included.

    *Code availability.* The code to construct a decision tree model in detecting the occurrence of blowing snow is available from GitHub (https://github.com/zpxie-cas/DTM).

**Technical comments**

**Text**

27. P2 L 64: the paper by Lehning et al (2000) coud also be mentioned in this list of references:

    Lehning, M., Doorschot, J., & Bartelt, P. (2000). A snowdrift index based on SNOWPACK model calculations. Annals of Glaciology, 31, 382-386.

    **Response:** This reference has been added to the list.

28. P 3 L 80:  Gilbert et al by should be replaced by Guyomarc'h et al.

    **Response:** Replaced.

29. P3L 91: wind direction can be removed from this list since it is not used by the authors.
    **Response:** Yes, wind direction has been removed.

30. P3L 91: the acronym "ISAW" should be defined here.
    **Response:** The ISAW is a product division of the IAV technology company, its full name is not available and the acronym "ISAW" is widely used in many references, for example, He and Ohara (2017), Vionnet et al (2013, 2018).

31. P 5 L 129: "wet precipitation" could be replaced by "liquid precipitation".
    **Response:** Yes, this phrase has been replaced thoroughly in this manuscript.

32. P 6 L 146: is the word "package" missing after "scikit-lean"?
    **Response:** The word has been added after "scikit-lean", thanks for pointing this out.

33. P 21 L 428: "microphysical" should be replaced by "microstructural".
    **Response:** The word has been replaced based on your suggestion.

**Figure**
34. Figure 1: This figure is really interesting. Can the author make the different subfigures slightly bigger?
    **Response:** This figure has been adjusted.

35. The upper graphs on figures 2, 3, 6, 7 are hard to read.
    **Response:** These figures have been adjusted

---

## Author Response (AR2)

The authors well addressed the reviwers' comments/suggestions and well revised the draft.

Only one small suggestion, please give the meaning of "BS" and "No\_BS" in the caption of figure 1. Are the abbreviation of "Blowing snow" and "No Blowing snow"? If this could be added, the manuscript could be final published.

**Response:** Thank you very much for your time to review our manuscript and providing so many valuable comments and suggestions. The following content has been added to describe the meaning of "BS" and "No\_BS".

BS and No\_BS denotes with and without blowing snow occurrence, respectively.

**Anonymous Referee #2**

The authors have carefully revised the manuscript based on the comments raised during the first round of review. Therefore, the manuscript had been made clearer and improved in many respects. I would like to thank the authors for this work. I have listed some minor and technical comments that should be addressed prior to publication.

**Response:** Thank you very much for your time to review our manuscript and providing so many valuable comments and suggestions. All the technical comments have been addressed.

The page and line numbers below refer to the revised version of the paper in Track Change mode.

- Specific comments:

P 2 L 71: The revised sentence proposed by the authors is somehow confusing. Indeed, it suggests that the SPC is an acoustic sensor, which is not the case. I recommend the authors to reformulate this sentence. Maybe: "..... the optical sensors deployed in the Antarctic and Alps (Snow Particle Counters, SPC; Sato et al., 1993; Nishimura and Nemoto, 2005; Vionnet et al., 2013), .... "

Response: Corrected.

P2 L 83-84: at which height above the snow surface are these values of the threshold wind speed valid?

**Response:** The following modification has been made to present the height of the threshold wind speed.

Threshold wind speed at the height of 10 m was found to be 9.9 m s-1 for wet snow and 7.7 m s-1 for dry snow, and a formula expresses the threshold wind speed as a function of air temperature has been proposed based on field observations from the Canadian Prairies (Li and Pomeroy, 1997a).

P 7 L 203-205 and Fig. 2 on P 6: it is not clear how is defined the frequency of blowing snow occurrence shown on this figure. Is it the ratio between the number of occurrences of blowing snow for a given atmospheric condition divided by the total number of

occurrences of this atmospheric condition?

**Response**: Yes. The following text has been added to the caption of Figure 2 to clearly state the blowing snow occurrence.

The blowing snow frequency denotes the ratio between occurrences of blowing snow for a given atmospheric condition divided by the total number of occurrences of this atmospheric condition.

P8 L 250-255: The authors should define the meaning of the acronym MR and should explain in the next paragraph what is measured by MR as already done for the other metrics (see P 9 L 270-278).

**Response**: Thanks very much for pointing this out. The following sentence has been modified to explain the meaning of MR.

The FAR measures the fraction of forecasted events that did not actually occur and the MR denotes the proportion of blowing snow events that actually occurred but not captured by the DTM model (both range from 0 to 1, with optimal performance of 0)

- Technical comments:

P 5 L 154 and throughout the text: consider using the notation m s^{-1} instead of m/s for the units of the wind speed. **Response**: Corrected.

P 9 Table 4: do the authors mean "dry snow cover" and "wet snow cover" instead of "dry snow covered" and "wet snow covered"? **Response**: Corrected.

P 22 L 647: the authors could refer here to Table 2 that describes the different tests

P 26 L 791: the sentence " ... with downscaled by SAFRAN ..." is not clear. I guess "with should be removed." **Response**: Corrected.